# Evaluating the Usability of mHealth Applications on Type 2 Diabetes Mellitus Using Various MCDM Methods

**DOI:** 10.3390/healthcare10010004

**Published:** 2021-12-21

**Authors:** Kamaldeep Gupta, Sharmistha Roy, Ramesh Chandra Poonia, Soumya Ranjan Nayak, Raghvendra Kumar, Khalid J. Alzahrani, Mrim M. Alnfiai, Fahd N. Al-Wesabi

**Affiliations:** 1Faculty of Computing and Information Technology, Usha Martin University, Ranchi 835103, India; kamal.sxcranchi@gmail.com (K.G.); sharmistha@umu.ac.in (S.R.); 2Department of Computer Science, CHRIST (Deemed to Be University), Bangalore 560029, India; rameshcpoonia@gmail.com; 3Amity School of Engineering and Technology, Amity University Uttar Pradesh, Noida 201301, India; nayak.soumya17@gmail.com; 4Department of Computer Science and Engineering, GIET University, Rayagada 765022, India; raghvendra@giet.edu; 5Department of Clinical Laboratories Sciences, College of Applied Medical Sciences, Taif University, P.O. Box 11099, Taif 21944, Saudi Arabia; Ak.jamaan@tu.edu.sa; 6Department of Information Technology, College of Computers and Information Technology, Taif University, P.O. Box 11099, Taif 21944, Saudi Arabia; m.alnofiee@tu.edu.sa; 7Department of Computer Science, College of Science & Art at Mahayil, King Khalid University, Abha 61421, Saudi Arabia; 8Department of Information Systems, Faculty of Computer and Information Technology, Sana’a University, Sanaa 1993, Yemen

**Keywords:** T2DM, mHealth applications, critic, usability score

## Abstract

The recent developments in the IT world have brought several changes in the medical industry. This research work focuses on few mHealth applications that work on the management of type 2 diabetes mellitus (T2DM) by the patients on their own. Looking into the present doctor-to-patient ratio in our country (1:1700 as per a Times of India report in 2021), it is very essential to develop self-management mHealth applications. Thus, there is a need to ensure simple and user-friendly mHealth applications to improve customer satisfaction. The goal of this study is to assess and appraise the usability and effectiveness of existing T2DM-focused mHealth applications. TOPSIS, VIKOR, and PROMETHEE II are three multi-criteria decision-making (MCDM) approaches considered in the proposed work for the evaluation of the usability of five existing T2DM mHealth applications, which include Glucose Buddy, mySugr, Diabetes: M, Blood Glucose Tracker, and OneTouch Reveal. The methodology used in the research work is a questionnaire-based evaluation that focuses on certain attributes and sub-attributes, identified based on the features of mHealth applications. CRITIC methodology is used for obtaining the attribute weights, which give the priority of the attributes. The resulting analysis signifies our proposed research by ranking the mHealth applications based on usability and customer satisfaction.

## 1. Introduction

The one of the most common chronic disorders that affect people today is diabetes mellitus, and its incidence is quickly increasing around the world (According to WHO report on diabetes (2021), it has been mentioned that, in 2014, 8.5% of adults aged 18 years and older had diabetes, and in 2019, diabetes was the direct cause of 1.5 million deaths, and 48% of all deaths due to diabetes occurred before the age of 70 years). [1,2]. Nowadays, we are living in the era of smart technology in healthcare domain. As a result of mobile technology such as smart phones, wearable devices, and other smart devices, the use of mHealth applications is growing at an exponential rate. Hence, most poor countries have also embraced the use of mobile health applications to improve the delivery of health services [3]. Establishing a specific treatment plan, taking medicines, and adhering to regular blood glucose (BG) monitoring and medical nutrition therapy are all critical steps in improving diabetes control [4]. However, usability testing can be used by researchers to uncover flaws in the products and systems that exist and gain ideas about how to improve them. The ease with which a product can be controlled by users to achieve a particular objective in terms of efficiency, effectiveness, and satisfaction is referred to as usability [5]. When it comes to T2DM mHealth apps, the main goal for users is to find reliable, accurate, and timely information with minimal effort. As a result, quality is a critical aspect for T2DM mHealth applications, which must be monitored in order to achieve the stated goal. T2DM mHealth applications, on the other hand, are built to deliver accurate, reliable, and adequate information with high efficiency and low effort. To do this, T2DM mHealth applications must be more user-friendly. The usability of five T2DM mHealth applications namely, Glucose Buddy, mySugr, Diabetes: M, Blood Glucose Tracker, and OneTouch Reveal, is evaluated in this paper. All five T2DM mHealth applications are part of the same platform and offer outstanding and widespread assistance to type 2 diabetic patients with diabetes self-management. As a result, T2DM mHealth applications should provide users with useful information that takes minimal effort and time. The usability of five T2DM mHealth applications that may meet quality criteria in a broad sense is evaluated in this paper.

A pilot analysis of the study was carried out for recognizing the usability attributes that should be included when evaluating usability, taking into account the various types of users who access the T2DM mHealth applications. For the assessment of the utility of T2DM mHealth applications, several criteria and sub-criteria should be met. All five T2DM mHealth applications are evaluated for usability in order to determine the most usable T2DM mHealth applications for achieving user expectations and demands.

From the perspective of the user, the best T2DM mHealth applications are determined by the total usability score for each T2DM mHealth application, which is in turn determined by the quality of the T2DM mHealth applications. MCDM approaches are used to make the selection process easier. Due to its capacity to use specific criteria to evaluate numerous alternatives, MCDM is an excellent methodology for evaluating and analyzing difficult and complicated real-world situations [6]. The multi-criteria decision-making (MCDM) algorithms described in this work include TOPSIS, VIKOR, and PROMETHEE II. Criteria importance through the inter-criteria correlation (CRITIC) methodology is applied in this study. For evaluating the usability of T2DM mHealth applications, the CRITIC methodology is first utilized in order to find objective weights of all related criteria. CRITIC can effectively differentiate the properties of individual criteria by analyzing their respective strengths and evaluating criteria weights based on the relationships between them. The best alternative was chosen using a combination of criteria and sub-criteria. A variety of usability attributes and sub-attributes were employed as criteria and sub-criteria in this study. The main purpose of this study is to determine which of the T2DM mHealth apps is the most useful among the five alternatives—Glucose Buddy, mySugr, Diabetes: M, Blood Glucose Tracker, and One Touch Reveal—based on the ranking assigned by the different MCDM methods adopted in this paper, which include TOPSIS, VIKOR, and PROMETHEE II. The major contributions of this study are as follows:Identifying T2DM mHealth applications’ attributes and sub-attributes.Collection of feedback using questionnaire-based evaluation. Questionnaires are framed based on attributes and sub-attributes using a pilot study and expert opinions.Obtain objective weights of all connected criteria using the CRITIC technique.Evaluate the usability of the mHealth applications using three popular MCDM methodologies like TOPSIS, VIKOR, and PROMETHEE II.Ranking the alternatives based on the obtained usability score.Comparing the ranking of the alternatives across the three MCDM methods to check the consistency of the results.

The following is the organization of this paper. Section 2 presents related works on usability evaluation and MCDM techniques. Then, the paradigm and methodologies proposed for the process of usability evaluation are presented in Section 3. The results and analysis are discussed in Section 4, and in Section 5 the work is concluded.

## 2. Related Works

For plant site selection challenges and problems, the fuzzy technique for order preference by similarity to ideal solution (TOPSIS) methodology was introduced by Chu [7]. Firstly, the decision makers’ ratings and weights are standardized to develop a scale that is comparable. For all alternative sites and for every rating that is normalized, the membership function is developed for each criterion. The ranking of the alternatives is determined by a closeness or proximity coefficient. Amiri et al. [8] employed the TOPSIS technique in conjunction with heuristics based on fuzzy goal programming to discover the ideal position. The challenge of selecting the facility location is handled using three stages: (a) choosing the smallest range of possible distribution centers, (b) situating those in the largest location that is feasible, and (c) locating these facilities at the lowest possible cost. Under a fuzzy context, Lo et al. [9] used TOPSIS to rank web services. They used five core criteria (cost, runtime, configuration management, transaction, and security). They also used 17 sub-criteria to choose the ideal web service based on the team’s performance. For evaluating and analyzing the overall performance of eHealth systems, Lohan et al. [10] developed a MCDM model. The eHealth efficiency was evaluated by authors using five criteria that include ease of use, cost, acceptability, false alarm, and accuracy.

Grigoroudis et al. [11] created a method for evaluating performance as well as a review system based on a balanced scorecard for public health care services. In order to preserve service quality, customer contentment, the organization’s self-improvement system, and the organization’s ability to adapt and evolve, both the financial and the non-financial performance metrics are included in this evaluation method. Researchers compared TOPSIS, VIKOR, and ELECTRE, the three multi-criteria decision-making (MCDM) methodologies, to see how well they could evaluate and assess the performance and the quality of 21 food companies [12]. A further, comparative analysis of AHP, TOPSIS, and PROMETHEE, [13] observed that certain decision-making approaches are still susceptible to confusion where a choice has to be made between two or more alternatives in relation to the subjectivity. Mahmoodzadeh et al. [14] used a hybrid of fuzzy AHP and the TOPSIS techniques in order to develop a project-selecting strategy that employs the enhanced technique for obtaining the attribute or criterion weights, and then for ranking the projects, the algorithm of TOPSIS was chosen.

Gavade [15] looked into a variety of multi-criteria decision-making situations. As examples of MCDM techniques for various cloud services, they studied AHP, TOPSIS, VIKOR, ELECTRE, and PROMETHEE. TOPSIS could be used for PaaS decision making, according to the authors. “The Secure use score to achieve system usability” was proposed by Peikari, H. R. et al. [16] in 2018. Every characteristic in this measure was given a score by the authors. However, only three usability parameters were examined for this metric, and no case study was used to evaluate it. In 2019, Hsieh, M.H. et al. [17] assessed the utility of the available three diabetic self-management smartphone applications. In this study, the usability of diabetes applications was also examined with the help of usability testing. The usability study involved a total of 30 participants having type 2 diabetes (15 men and 15 women). The participants’ average age was 60.03 years, having a standard deviation of 8.92 years. Jayant. A. et al. [18] classified major works involving MCDM methodologies that include PROMETHEE, VIKOR, ELECTRE, and TOPSIS, and provided help in identifying the literature and future research opportunities. PROMETHEE, VIKOR, ELECTRE, and TOPSIS are all MCDM strategies that are studied and implemented in this research. Wang, F.K. et al. [19] analyzed and enhanced six sigma projects for minimizing overall performance and quality shortfalls within every criterion and parameter, and created a hybrid MCDM model that included the DEMATEL technique, the VIKOR method, and the analytic network process (ANP).

Ghaleb, A.M. et al. [20] established an approach for evaluating and comparing three different selection methods: the technique for order of preference by similarity to ideal solution (TOPSIS), VIKOR (stepwise procedure), and the analytic hierarchy process (AHP). This review procedure took into account the number of distinct techniques and criteria, efficiency in the decision-making procedure, computing difficulty, and sufficiency for the support of a group choice, and criterion of inclusion or deletion. This work included a case study to examine the evaluation process. The criteria or attributes employed to examine and selecting the best or the finest manufacturing process included accuracy, flexibility, quality, productivity, complexity, operation cost, and material use.

Wu. Z. et al. [21] compared and evaluated four multi-criteria decision-making methods including the analytic hierarchy process (AHP), elimination et choix traduisant la realité (ELECTRE III), the technique for order of preference by similarity to ideal solution (TOPSIS), and Preference Ranking Organization Methods for Enrichment Evaluations (PROMETHEE II) for one sewer network decision made by the group in the early stages for sewer water infrastructure’s asset management. Furthermore, the Delphi methodology is used to manage and organize talks across all decision makers during the implementation of numerous MCDM approaches. PROMETHEE II is the most popular approach among decision makers; AHP takes much time and effort, and there may be several irregularities as a result; while vector normalization for multi-dimension criteria is being done, TOPSIS loses the information, and the results of ELECTRE III are unclear. Bratati et al. [22] did a study related to the usability measures and various evaluation methodologies that aid in user satisfaction. The authors also explored a number of issues concerning usability and the need for usability models in the field of cloud computing. A survey was conducted by Roy and Pattnaik [23] to evaluate various usability criteria and different evaluation methods used to assess the usability and acceptability of web apps and websites. Based on the relevance of the online internet facility for all sorts of users, the authors recommended two new measures to demonstrate the usefulness of websites, including device independence and assistance for people who are physically challenged.

Liew et al. [24] in their research work conducted a qualitative study to provide a deep understanding of the usability parameter of different mHealth applications and also incorporate suggestions for upgrading the experience of users in terms of usability aspect. The authors explore the various alignments between the users and mHealth practitioners for conducting the study. It was observed that, based on the five major themes selected from 20 different applications, satisfaction is the top-ranked attribute, whereas intuitiveness was least preferred by the users.

Zhou et al. [25] and other authors have conducted a research study on developing a reliable usability questionnaire, especially for evaluating the usability of mHealth app popularly known as MAUQ, which has three subscales and is tested in both standalone as well as interactive mHealth applications. The obtained responses were compared with the standard Post Study System Usability Questionnaire (PSSUQ) as well as System Usability Scale (SUS), which seem to be correlated with each other.

Islam et al. [26] used a three-stage approach. The first stage was to do a keyword-based app search on the most popular app stores. The affinity diagram method was used, and the applications discovered were divided into nine groups. Secondly, four apps were chosen at random from each group (a total of 36 apps) and a heuristic evaluation was performed. Finally, in the third stage, the most downloaded app from each group was chosen, and user studies with 30 people were undertaken.

Isaković et al. [27] studied the diabetes monitoring app named DeStress Assistant (DeSA), which was created as part of an EU project and evaluated in a hospital context. An assessment of an available diabetes app was carried out in two test trials with older users, utilizing various questionnaires. Since the number of older persons is rising, the app is designed with their population in mind. The app, which was built with the support of workshops and comments from tech-savvy patients and healthcare professionals, is challenging to use by elderly users, according to a number of supervised tests.

Georgsson et al. [28] conducted a study to see if a multi-method technique of data collection and analysis for patients’ experience with a mobile health system for diabetes type 2 diabetes self-management is feasible. From a wider clinical trial, a random sample of 10 users was chosen. User testing involving eight typical tasks and the Think Aloud protocol, a semi-structured interview, and a questionnaire on patients’ experiences with the system were all used to obtain data. The results were structured, coded, and evaluated using the framework analysis (FA) approach and the usability problem taxonomy (UPT). After classification, a usability severity rating was applied.

Eberle et al. [29] in their study reviewed the clinical effectiveness of mHealth apps in managing diabetes mellitus patients of type 1 (T1DM), type 2 (T2DM), and gestational DM. A system review was conducted from a literature review carried out between January 2008 and October 2020, which was categorized based on the type of DM and results obtained. A meta-analysis report was prepared to measure the impact of glycated hemoglobin (HbA1c) on the different types of diabetes mellitus mHealth apps.

Teng et al. [30] studied how various authentication methods affect the usability of mHealth apps. Secondly, new metrics for evaluating ease of use were introduced, and thirdly, the usability of two prevalent authentication systems for mHealth apps was evaluated using numerous key process features and their influence on users. Based on the findings, a QR-code-based authentication method for mHealth apps was proposed, which would help users to overcome frequent barriers.

A systematic review of diabetes management apps for the iOS platform is described by Martin et al. [31]. The KLM review revealed several usability difficulties related to data entry and personalized settings, while the heuristic evaluation revealed additional issues related to devising loss, aesthetics, learn ability, error management, and security.

Timurtas and Polat [32] conducted a study to perform a comparison of the usability parameter of smartphone and smart watch devices that aims to help users suffering from type 2 diabetes as well as clinicians focusing on T2DM. Usability was measured by using the System Usability Scale (SUS), and a t-test was conducted to compare the scores obtained by SUS for both devices. It was observed that the usability of smartphone devices is higher when compared with smart watch devices, but overall both the devices have high usability scores (SUS score > 80.8) measured by users as well clinicians.

## 3. Proposed Methodology

Usability is defined as “the extent to which a product can be used by specified users to achieve specified goals with effectiveness, efficiency, and satisfaction in a specified context of use” [ISO, ISO 9241-11] [33]. According to Jakob Nielsen (1993) [34], “Usability is the measure of the quality of the user experience when interacting with something—whether a website, a traditional software application, or any other device the user can operate in some way or another.” In this study, usability measures the simplicity and ease of use with which users can utilize the T2DM mHealth applications to meet their needs and satisfaction. A questionnaire-based approach is used in this study to evaluate usability. The flowchart of the proposed methodology is shown in Figure 1 below.

### 3.1. Evaluation Based on Questionnaire

Participants respond to a standard questionnaire created by experts to assess the overall usability of the five T2DM mHealth applications. The pilot study’s goal is to find out and provide significant insights into the evaluation process. It assists in determining the characteristics and appropriate questions for conducting usability testing. In this case, different aspects like “Learnability”, “Efficiency”, “Memorability”, “Aesthetic”, “Error”, “Navigation”, “Readability”, “Cognitive Load”, “Provision for Physically Challenged Users”, and “Satisfaction” are the 10 usability attributes used to create the questionnaires. Each attribute has multiple sub-attributes that are necessary for assessing the usefulness of the five T2DM mHealth applications. For the 10 different usability attributes, 29 questionnaires were prepared [Appendix A]. The questions are graded on a linguistic rating scale of 1–5, in which 5 denotes strongly agree and 1 denotes strongly disagree.

#### Identifying the Usability Attributes and Sub-Attributes

The attributes and sub-attributes, which are evaluated for the usability of T2DM mHealth applications, are listed for providing clear insight and clarity, thereby making it convenient for participants to evaluate the five T2DM mHealth applications. The attributes defined for evaluating the usability of mHealth applications are identified based on numerous literature surveys as well as a perspective-based user interface inspection. In this study, perspective is gathered from the user profiles, their technical ability, and most important the medical practitioners who facilitate the mHealth applications by rendering different services and support to the patients for regular health monitoring related to T2DM. Various tasks were also carried out by the inspectors to explore the mHealth applications with a focus on the different perspectives. Table 1 below shows the attributes (criteria) and their corresponding sub-attributes (sub-criteria). Table 2 provides a briefing of the sub-attributes.

In this study, beneficial attributes include ‘Learnability’, ‘Efficiency’, ‘Memorability’, ‘Aesthetic’, Readability’, ‘Provision for Physically Challenged Users’, and ‘Satisfaction’, whereas non-beneficial attributes include ‘Error’, ‘Navigation’, and ‘Cognitive Load’. In the case of beneficial attributes, a score of 5 on the Likert Scale represents the maximum level of user satisfaction, while a value of 1 represents the minimum level. Whereas, in the case of non-beneficial attributes, 1 represents the highest level of user satisfaction and 5 represents the lowest level. Each sub-attribute is assessed using a standard questionnaire that is graded on a linguistic scale. The different qualities and sub-attributes employed in this investigation are depicted in Figure 2. For each of the attributes, scores are obtained using the linguistic scale. The feedback is provided by 30 users from various age groups who use the T2DM mHealth applications and will benefit from them.

### 3.2. Alternatives Chosen for Usability Evaluation

Alternatives chosen for the study are one of the best mHealth applications for monitoring diabetes patients. The screenshots of the applications are shown in Figure 3, Figure 4, Figure 5, Figure 6 and Figure 7. The services offered by these applications are mentioned below.

Glucose Buddy (Alt1): services offered by this application are as follows:Simple and hassle-free solution for controlling diabetes with real-time blood sugar measurement.Provides professional support and advice.

mySugr (Alt 2): the best features of mySugr application are as follows:Personalized logging screen, which can record the value from bluetooth-enabled blood glucose meter and analyze the pattern to brief the blood glucose levels.Smart search facility for recording meals and activities, which helps in controlling diabetes.Ability to provide the highest quality security as per the General Data Protection Regulation (GDPR).

Diabetes: M (Alt 3): this mHealth application provides everything needed for effective health management by offering the following services:Provides detailed information of user.Effective diabetes control remotely.Present the report in statistical format (like bar chart), which produces better understanding among the users.Feature of recognizing the pattern and look for any predefined recurring problems along with the reason for occurrence.Adds an insulin bolus calculator for calculating insulin based on nutritional information.

The application Blood Glucose Tracker (Alt 4): the services offered by this application are as follows:Tracking blood glucose at every level (like breakfast, lunch, and dinner) throughout the day, thereby helping patients to control blood sugar efficiently.Moreover, it can also record blood pressure, weight, HbA1c, etc.

OneTouch Reveal (Alt 5): the unique features of this mHealth application are as follows:It provides unique color-coding technology to organize blood sugar results, which can be easily understood by naïve users.It automatically notifies repeated highs or lows so that proper action can be taken.This alternative also sets the goal for recording steps walked daily, carbs, and activity.It set reminders for undergoing blood sugar test as well as for taking insulin.

### 3.3. Evaluating Usability Using MCDM Methods

The multi-criteria decision-making (MCDM) method is amongst the most widely used decision-making techniques in the fields related to science, government, business, and engineering. By improving the decision-making process and making it more transparent, efficient, and logical, MCDM techniques can enhance the quality and effectiveness of making decisions. Choices are made based on hierarchical comparisons of multiple options that are frequently dependent on conflicting criteria throughout the decision-making process. These decisions should be made using multi-criteria decision-making (MCDM) methodologies. MCDM methodologies are widely employed in solving problems associated with multiple, inconsistent, and disproportionate objectives and/or criteria. The MCDM procedures are used to break down complicated problems into minor sections [35], so that after the analyses are completed, all of the parts may be put together to provide a holistic picture of the associated problem. MCDM techniques enable the decision maker to consider a variety of criteria or objectives in an effort to find a middle ground between possibly conflicting criteria or variables [36,37]. As a result, the decision maker should assess and evaluate both qualitative and quantitative criteria [38].

#### 3.3.1. CRITIC Method

In generating objective weights, the criteria importance through inter-criteria correlation (CRITIC) methodology takes into consideration the standard deviation of data for each criterion as well as the correlation between the criteria. Diakoulaki et al. [39] introduced the CRITIC technique in the year 1995. It is employed for finding the objective weights of all connected criteria while evaluating T2DM mHealth apps. CRITIC can effectively differentiate the characteristics of individual criteria by analyzing their respective strengths and evaluating criteria weights based on the relationships between them. Figure 8 depicts the steps involved in the CRITIC technique.

Description of the method: To begin, the problem is considered to have a set or collection of *m* feasible alternatives *A**_i_*(*i* = 1, 2, …, *m*) and *n* criteria for evaluation.

Step 1: Create a decision matrix.

The following depicts the different alternatives’ performances based on multiple attributes or criteria.
$$X={\left[x_{ij}\right]}_{m\times n}=\left[\begin{array}{cccc} x_{11} & x_{12} & \ldots & x_{1n}\\ x_{21} & x_{22} & \ldots & x_{2n}\\ : & : & : & :\\ x_{m1} & x_{m2} & \ldots & x_{mn} \end{array}\right]\left(i=1,2,\ldots ,m\;and\;j=1,2,\ldots ,n\right)$$
where $x_{ij\;}$ represents performance value related to the *i*^th^ alternative with respect to the *j*^th^ criterion.

Step 2: Normalize the decision matrix.

First, determine the best and worst values in relation to beneficial and non-beneficial attributes or criteria, respectively. In the case of beneficial criteria, the maximum value will be the best value, whereas the minimum value will be the best preferable value in regard to non-beneficial criteria.

Now, the matrix is normalized as follows:(1)$$x_{ij}^{*}=\frac{x_{ij}-min\left(x_{ij}\right)}{max\left(x_{ij}\right)-min\left(x_{ij}\right)}\;\;i=1,2,\ldots m\;and\;j=1,2,\ldots ,n$$
where $x_{ij}^{*}$ is the performance value that is normalized related to the *i*^th^ alternative with respect to the *j*^th^ criterion. The type of criteria is not taken into consideration in the case of normalization.

Step 3: Determine the standard deviation and correlation coefficient. 

The standard deviation of each criterion is determined. $\sigma_{j}$ is the standard deviation related to the *j*^th^ criterion. Then, for each pair of criteria, the distance related to correlation is determined. $r_{jk}$ is the correlation coefficient between the two attributes or criteria.

Step 4: Obtain the quantity of information.

The quantity or volume of information included in the *j*^th^ criterion is denoted by *C**_j_* and is obtained as:(2)$$C_{j}=\sigma_{j}\cdot \overset{m}{\underset{k=1}{\sum }}(1-r_{jk})$$


Step 5: Calculate and obtain the criteria weights/

The weight related to the *j*^th^ criterion (*w_j_*) is calculated as:(3)$$w_{j}=\frac{C_{j}}{\sum_{k=1}^{m}C_{k}}$$

This method, as stated, gives a higher weight to attributes or criteria with a significantly high value of standard deviation and low value of correlation in comparison to all other criteria [40]. Specifically, a high value of *C_j_* indicates the maximum amount of information yielded by the provided criterion, implying that the importance of criterion for the problem of decision making is greater.

#### 3.3.2. TOPSIS

Hwang and Yoon first introduced the TOPSIS methodology in 1981 [41], which falls under the category of aspiration, goal, or reference-level approaches. According to the basic principle of this method, the optimal or efficient alternative is closest to the ideal or best solution and furthest away from the anti-ideal or worst option. The best or ideal solution assists in maximizing the benefit requirements or criteria and minimizing the cost criteria. In the case of anti-ideal or worst solution, on the other hand, the cost criterion is maximized while the benefit criterion is minimized. 

The TOPSIS approach is depicted in Figure 9 as a series of steps. Following is the description of the procedures or the steps.

Step 1: Generate the decision matrix.

$A={\left[x_{ij}\right]}_{m\times n}$ is a matrix that contains m alternatives, denoted as *a*_1_, *a*_2_, …, *a_m_*, and n criteria, which are represented as *C*_1_, *C*_2_, *C*_3_, …, *C**_n_* with *x**_ij_* representing the intersection of each choice and characteristic or criterion, and it is generated as shown below:$$X={\left[x_{ij}\right]}_{m\times n}=\left[\begin{array}{cccc} x_{11} & x_{12} & \ldots & x_{1n}\\ x_{21} & x_{22} & \ldots & x_{2n}\\ : & : & : & :\\ x_{m1} & x_{m2} & \ldots & x_{mn} \end{array}\right]\left(i=1,2,\ldots ,m\;and\;j=1,2,\ldots ,n\right)$$

Step 2: Calculate the matrix R = ${\left(r_{ij}\right)}_{m\times n}$ that is standardized by the following equation.
(4)$$r_{ij}=\frac{a_{ij}}{\sqrt{\overset{m}{\underset{k=1}{\sum }}}=a_{kj}^{2}},i=1,2,\ldots ,m,j=1,2,\ldots n$$

Step 3: The weighted standardized matrix is determined.
(5)$$T={\left(t_{ij}\right)}_{m\times n}\;t_{ij}=r_{ij}.\omega_{j},i=1,2,\ldots ,m,j=1,2,\ldots ,n$$
where $\omega_{1},\;\omega_{2}\;$,… are the weights which are associated with the criteria and $\overset{n}{\underset{j=1}{\sum }}\omega_{j}=1$

Step 4: The ideal (best) solution and anti-ideal (worst) solution are found.

The ideal solution is denoted by $S^{+}$ and the anti-ideal solution is signified as $S^{-}$ and are given as follows:$$S^{+}=\left\{t_{j}^{+}|j=1,2,\ldots ,n\right\}=\left\{\left(\underset{i}{min\;}t_{ij}|j\in J^{-}\right),\left(\underset{i}{max\;}t_{ij}|j\in J^{+}\right)\right\}$$
$$S^{-}=\left\{t_{j}^{-}|j=1,2,\ldots ,n\right\}=\left\{\left(\underset{i}{max\;}t_{ij}|j\in J^{-}\right),\left(\underset{i}{min\;}t_{ij}|j\in J^{+}\right)\right\}$$
where $J^{+}$ represents the beneficial and $J^{-}$ denotes the non-beneficial criteria, respectively.

Step 5: Compute Euclidean distance.

Then, in the following equations, we compute the Euclidean distance having n-dimensions between the alternative i and the ideal (best) solution $S^{+}$ as well as the anti-ideal (worst) solution $S^{-}$, represented as $D_{j}^{+}$ and $D_{j}^{-}$, respectively:(6)$$D_{i}^{+}=\sqrt{\overset{n}{\underset{j=1}{\sum }}{\left(t_{ij}-t_{j}^{+}\right)}^{2}}$$
(7)$$D_{j}^{-}=\sqrt{\overset{n}{\underset{j=1}{\sum }}{\left(t_{ij}-t_{j}^{-}\right)}^{2}}$$

Step 6: Measure relative closeness.

The alternative’s relative proximity or closeness to the ideal (best) solution is measured by the following:(8)$$C_{i}=\frac{D_{i}^{-}}{\left(D_{i}^{+}+D_{i}^{-}\right)}$$

Step 7: Rank the alternatives.

Finally, based on the *C_i_* values, we assign a score or rating to each alternative. The maximum value to the problem is the ideal solution. 

#### 3.3.3. VIKOR

Opricovic [42] introduced the VIKOR technique, which has now become a well-known MCDM strategy centered on choosing and ranking the alternative sets of conflicting criteria. In recent years, scholars have become increasingly interested in this technique. The Vlse Kriterijumska Optimizacija I Kompromisno Resenje (VIKOR) technique, identified as a flexible ranking strategy for acquiring the optimum decision-making steps or procedures [43], is one of the MCDM approaches. In the presence of contradictions, the VIKOR technique is implemented by rating and selecting among a set of alternatives [44]. The optimal option is determined using the MCDM compromise approach, which ranks the alternatives according to how near they are to the ideal (best) solution. The VIKOR concept was introduced as a potential MCDM technique [42]. A feasible solution is regarded as a compromise solution if it comes very close to the ideal (best) solution, whereas a compromise is a mutually agreed-upon agreement.

The basic procedures of the VIKOR technique are shown in Figure 10 and exhibited as follows.

Step 1: Linguistic variables values are provided that are relevant to processing the alternatives. 

Following that, a matrix related to alternatives for each criterion must be created.

Step 2: A decision matrix is generated.

Expert scores and ratings are used to collect alternative ratings, and the following equation is used to create a decision matrix:(9)$$A=\frac{1}{k}\overset{n}{\underset{i=1}{\sum }}A_{ij},\;\mathrm{for}\mathrm{all}\;j=1,2,\;\ldots ,\;m$$
where the total number of decision makers is denoted by k, i represents the total amount of alternatives, and the total amount of criteria is j.

Step 3: Identification of best and worst values of all attribute or criterion functions.

Find the best $f_{b}^{*}$ and the worst $f_{b}^{-}$ ratings of the criterion, where b = 1,2,…, n. The best $f_{b}^{*}$ and worst $f_{b}^{-}$ values are calculated using the equations given below:(10)$$\;\;\;\;\;\;\;\;\;\;f_{b}^{*}=max\left(f_{ab}\right)\;\mathrm{and}\;f_{b}^{-}=min\left(f_{ab}\right)$$
where $f_{b}^{*}$ denotes the positive ideal solution (PIS) related to the b^th^ criterion. The negative ideal solution (NIS) related to b^th^ criterion is given by $f_{b}^{-}$.

Step 4: Calculate the unity and regret measures.

Compute $S_{a}$ (unity measure) as well as the value $R_{a}$ (regret measure) for a= 1,2,…, n with the help of the equations that follow:(11)$$S_{a}=\overset{n}{\underset{b=1}{\sum }}W_{b}\left[\frac{f_{b}^{*}-f_{ab}}{f_{b}^{*}-f_{b}^{-}}\right],$$
(12)$$R_{a}=max_{b}[W_{b}[\frac{f_{b}^{*}-f_{ab}}{f_{b}^{*}-f_{b}^{-}}]],$$
where $S_{a}$ means the distance rate to the PIS, i.e., highest “group utility of majority,” $R_{a}$ signifies the distance rate to NIS, i.e., lowest “individual regret of the opponent,” and $W_{b}$ specifies the weight of each criterion determined using the CRITIC technique.

Step 5: Find the highest (maximum) and smallest (minimum) values from the utility and the regret measures.
$$S^{-}=max_{a}S_{a}\;S^{*}=min_{a}S_{a}\;R^{-}=max_{a}R_{a}\;R^{*}=min_{a}R_{a}$$

Step 6: Calculate the final negotiating solution, which is the aggregating index $Q_{a}$ for a=1,2,…,m.

The best preferred alternative has the lowest value of $Q_{a}$.
(13)$$Q_{a}=v\frac{S_{a}-S^{*}}{S^{-}-S^{*}}+\left(1-v\right)\frac{R_{a}-R^{*}}{R^{-}-R^{*}},$$

The solutions that accomplish $max_{a}S_{a}$ denote the “largest (maximum) group majority”, whereas the solutions that achieve $min_{a}S_{a}$ denote the “minimum (least) individual regret” for the alternative. v represents the action path’s weight or maximum utility of the set. The weight of the individual contribution is denoted by (1 − v). In this investigation, 0.5 is taken as the value of v.

Step 7: Rank the alternatives.

Rank the alternatives based on their $Q_{a}$ scores.

If the following two conditions are met, the best negotiation or compromise solution will have a minimum value of Q and can be suggested:(1)If $Q\left(A^{\left(2\right)}\right)-Q\left(A^{\left(1\right)}\right)\geq 1/n-1$, alternative $Q\left(A^{\left(2\right)}\right)$ indicates a useful feature. Here *n* represents the total size of alternatives, whereas $Q\left(A^{\left(2\right)}\right)$ and $Q\left(A^{\left(1\right)}\right)$ represent the alternatives.(2)If $Q\left(A^{\left(1\right)}\right)$ is identified as the best in $S_{a}$ and $R_{a}$, it is stable and consistent in the process of making the decisions. The best or optimal alternative is $\left(A^{\left(m\right)}\right)$, which has the least value of $Q_{a}$ in terms of the conditions listed above, where “m” signifies the number of choices or alternatives.

#### 3.3.4. PROMETHEE II

One frequently used MCDM method is the PROMETHEE method [45]. It has the ability to rank the decision options. To rank the alternatives, it calculates the positive and negative outranking flow. Brans [46] originally released PROMETHEE I and PROMETHEE in 1982. Only partial ranking of the options or alternatives is created by PROMETHEE I, whereas PROMETHEE II can generate a complete rank for the alternatives. In this paper, the PROMETHEE II approach is used to obtain a complete ranking of T2DM mHealth applications taken into consideration. Figure 11 depicts the steps involved in PROMETHEE II.

The PROMETHEE II method’s procedural steps are listed below.

Step 1: Use the following equation to normalize the decision matrix:(14)$$R_{ij}=\frac{\left[X_{ij}-min\left(X_{ij}\right)\right]}{\left[max\left(X_{ij}\right)-min\left(X_{ij}\right)\right]}\left(i=1,2,\ldots ,n:j=1\ldots \ldots ,m\right)$$
where $X_{ij}$ indicates the performance of the *i*^th^ alternative in relation to the j^th^ criterion. When it comes to the criterion that is non-beneficial, Equation (1) can be rewritten in the following way:(15)$$R_{ij}=\frac{\left[max\left(X_{ij}\right)-X_{ij}\right]}{\left[max\left(X_{ij}\right)-min\left(X_{ij}\right)\right]}$$

Step 2: The i^th^ alternative’s evaluative differences from the other alternatives are determined. This stage entails computing the differences in the values of criteria between pairs of options or alternatives.

Step 3: Determine the $P_{j}\left(i.i^{\prime }\right)$ function, which is the preference function.

According to Brans and Mareschal [46], there are six different forms of generalized preference functions. For these forms of preference functions to work; however, other preferred parameters, which include the preference and the indifference thresholds, must be defined. However, in practical scenarios, determining which precise preference function form is acceptable with each criterion, as well as the parameters that are involved, maybe problematic for the decision maker. To prevent this issue, we use the simplified preference function as below:(16)$$P_{j}\left(i.i^{\prime }\right)=0\;\;if\;R_{ij}\leq R_{i^{\prime }j}$$
(17)$$P_{j}\left(i.i^{\prime }\right)=\left(R_{ij}-R_{i^{\prime }j}\right)\;\;if\;R_{ij}\leq R_{i^{\prime }j}$$

Step 4: Calculate the aggregated preference function while taking the criterion weights into account.

Computation of aggregated preference function can be done as follows,
(18)$$\pi \left(i^{\prime }.i\right)=\left[\overset{m}{\underset{j=1}{\sum }}W_{j}xP_{j}\left(i.i^{\prime }\right)\right]/\overset{m}{\underset{j=1}{\sum }}W_{j}$$

In this,$\;W_{j}$ denotes the weight of relative importance in relation to the j^th^ criterion.

Step 5: Leaving and the entering outranking flows are determined below.

For i^th^ alternative, positive or leaving flow is given as,
(19)$$\varphi^{+}\left(i\right)=\frac{1}{n-1}\overset{n}{\underset{i^{\prime }=1}{\sum }}\pi \left(i^{\prime }.i\right)$$

For i^th^ alternative, negative or entering flow is given as,
(20)$$\varphi^{-}\left(i\right)=\frac{1}{n-1}\overset{n}{\underset{i^{\prime }=1}{\sum }}\pi \left(i^{\prime }.i\right)$$

Here, the number of alternatives is given by n.

There are n−1 other alternatives surrounding every option. The outgoing (leaving) flow depicts how dominant one alternative is over others, and the incoming (entering) flow depicts how dominant another alternative is over others. The PROMETHEE I approach gives a partial preordering of the options or alternatives based on these outranking flows. The PROMETHEE II method, on the other hand, can provide a complete preorder using a net flow but loses a great deal of information on preference comparisons.

Step 6: Calculate each alternative’s net outranking flow.
(21)$$\varphi \left(i\right)=\varphi^{+}\left(i\right)-\varphi^{-}\left(i\right)$$

Step 7: Based on the values of φ(i), determine the order in which all of the alternatives (options) are ranked.

A greater value of (i) depicts that the alternative is better. As a result, the best option is the one with the greatest φ(i) value.

## 4. Result Analysis

A In this section, we analyze examples according to the computation steps from the previous section.

### 4.1. Participants

The usability testing of T2DM self-care mobile applications included 30 people from the Jharkhand region. Glucose Buddy, mySugr, Diabetes: M, Blood Glucose Tracker, and OneTouch Reveal were among the T2DM applications tested by the participants. There were 10 criteria in total, with 29 sub-criteria. The age of the participants was in the range from 41 to 77 years, with 15 women and 15 men. The participants’ average age was 59.33 years. All of the participants were diagnosed having type 2 diabetes mellitus and used smartphones and tablets frequently. The demographics of the participants are given in Table 3.

### 4.2. Measure Usability Score

Users’ feedback is taken based on the 5-point Likert Scale. Table 4 shows the usability score of each attribute.

### 4.3. Obtain the Objective Weights

The CRITIC technique was employed to obtain objective weights related to the 10 criteria or attributes. The steps in Section 3.3.1. were used to calculate the weights of both the beneficial and non-beneficial criteria (attributes), which are then listed in Table 5. Figure 12 compares the weights of the different attributes.

### 4.4. Ranking the Alternatives

After calculating the objective weights for each attribute, the next phase is to rank the alternatives using MCDM approaches like TOPSIS, VIKOR, and PROMETHEE II.

#### 4.4.1. Result Based on TOPSIS

The ranking was assigned to all the five alternatives i.e., Glucose Buddy, mySugr, Diabetes: M, Blood Glucose Tracker, and OneTouch Reveal by undergoing the steps mentioned in Section 3.3.2. The relative proximity or closeness to optimum or ideal solution ($C_{i}$) and the performance measure were found. The performance measure’s range or relative closeness of the T2DM applications discussed in this study between 0.271650464 and 0.748031419. Using the values of $C_{i}$, the ranking is then assigned. The option or alternative having the maximum of $C_{i}$ is the best solution and will be assigned Rank 1. The ranks assigned using the values of $C_{i}$ in TOPSIS method are shown in Table 6. Figure 13 represents the ranks obtained by the different alternatives. Thus, based on TOPSIS method, mySugr is the best T2DM mHealth application among the five alternatives.

#### 4.4.2. Result Based on VIKOR

All the five alternatives undergo the calculation process as given in the steps as mentioned in Section 3.3.3. The best alternative has the least (minimum) value with respect to the aggregating index ($Q_{a}$) and can only be suggested if conditions 1 and 2, as specified in Section 3.3.3, are satisfied. The T2DM applications’ aggregating index, or final negotiation solution, ranges between 0 and 0.898883249. The ranking assigned to the applications based on the values of $Q_{a}$ is depicted in Table 7, and the rank assigned to different alternatives is shown in Figure 14. Thus, based on the VIKOR method, mySugr is the best application and Blood Glucose Tracker is the least accepted application among the available five applications.

#### 4.4.3. Result Based on PROMETHEE II

One of the most commonly used MCDM methods for determining the rank is PROMETHEE II. All the five applications considered in this study underwent the calculation process as given by the steps of Section 3.3.4. Based on the value of net outranking flow, i.e., *φ*(*i*), ranking is assigned to applications. The net outranking flow of the T2DM applications is in the range between −0.26391893 and 0.344439094. A maximum (higher) value of *φ*(*i*) denotes the best alternative. Table 8 illustrates the PROMETHEE II method’s ranking of the alternatives based on the values of *φ*(*i*). Figure 15 shows the alternatives’ ranking using the PROMETHEE II method based on the values of *φ*(*i*). Thus, we can say that based on PROMETHEE II that mySugr is the best application among the five applications considered in this study, whereas Blood Glucose Tracker is the least accepted among the users.

#### 4.4.4. Comparison among the Different Alternatives Based on the Ranking Depicted by TOPSIS, VIKOR, and PROMETHEE II

As depicted in Table 9, based on the result of all the MCDM techniques (TOPSIS, VIKOR, and PROMETHEE II), mySugr is the best ranked T2DM mHealth application among the other applications considered in this study, whereas Blood Glucose Tracker is the least preferred application among the users. Here, 1 indicates the best ranked app and 5 indicates the least ranked app among all the alternatives. The ranking order of the T2DM mHealth applications considered in this study was the same for TOPSIS and PROMETHEE II methods. In VIKOR, there was a slight difference in the ranking order of Diabetes: M and Glucose Buddy. Diabetes: M had Rank 3, whereas Glucose Buddy has Rank 4. The comparison result of these three alternatives is depicted in Figure 16 below.

#### 4.4.5. Discussion and Comparison of the Proposed Methodology

The proposed research work is focused on evaluating the usability of mHealth applications, which became an essential tool in the present scenario of COVID-19. User experience while accessing the mHealth applications for day-to-day monitoring of chronic diseases like diabetes mellitus plays a vital role, because the efficiency and effectiveness of the mHealth applications ultimately improve user satisfaction. In the present study, the authors evaluated the usability of five mHealth applications that are purely utilized for controlling diabetes. The purpose of this study is not only limited to usability evaluation, but also to the selection of the best mHealth application among the alternatives available. For this selection purpose, a ranking was obtained for the five alternatives using the three most popular MCDM algorithms, namely TOPSIS, VIKOR, and PROMETHEE II. The proposed study selected these three MCDM models based on a research work conducted by Salabun et al. [47] where a comparative analysis was carried among several multi-criteria decision analysis (MCDA) methods, to benchmark some selected methods, namely TOPSIS, VIKOR, PROMETHEE II, and COPRAS. The analysis was performed based on various weighing methods (such as entropy and standard deviation) and with various techniques of normalization of MCDA model input data. It as observed from the result that, despite several MCDA methods available, no single method is perfect and cannot be considered suitable for any decision-making problem. This is because not only the methods themselves, but also the combination of normalization and other parameters, produce different results. Results also vary with the increase in the number of alternatives. The research work conducted aims in bringing out the similarity of rankings obtained using the four selected methods. It is observed from the study that TOPSIS, VIKOR, PROMETHEE II, and COPRAS are considered the best and most popular MCDA methods and are suitable in diverse domains like sustainability assessment, healthcare management, supplier selection, environment management, and many more.

The results obtained from the study also signify that the ranking obtained varies depending on the weights and normalization factor. It was observed that the ranking obtained using VIKOR was less correlated as compared to ranking obtained by the other three methods. Hence, the authors in this manuscript decided to check the consistency of the results obtained by the three MCDM methods.

Comparing the proposed study with other research works, certain differences were observed, which are listed below:Evaluating the usability of mHealth applications based on 10 criteria and 29 sub-criteria surely helps in accessing the user interface in a detailed manner. The obtained usability score for each criterion and sub-criterion helps in identifying the feature that hinders the overall usability of the applications.The MCDM models, such as TOPSIS, VIKOR, and PROMETHEE II, have a better decision-making ability for conflicting as well as non-conflicting criteria.A consistency check was performed on the results obtained from the three MCDM models, which signifies that all the models are correlated.

## 5. Conclusions

As we all know, the present doctor-to-patient ratio in our country is low, so it is very much essential to digitalize the health services. The only way to do so is through mHealth applications, which have become the medium for delivering health services in real time and also for facilitating consultation from remote locations. The use of mHealth applications for self-management of T2DM patients has saved various lives and has become the need of the modern world. However, all these facilities require efficient mHealth application with good user interface design for improving task effectiveness and user satisfaction. Thus, usability evaluation will surely help in identifying the best mHealth applications, which can, in turn, improve the health outcomes by improving patient experience of care and facilities provided, saving cost, and time while visiting a doctor’s office, facilitating real-time health monitoring, and other benefits. Hence, improving the usability and efficiency of the mHealth applications is essential for better customer satisfaction and trust. This research work identifies 10 criteria (attributes) and 29 sub-criteria (sub-attributes) based on the features of mHealth applications and expert opinions, which are used in questionnaire-based evaluations. For measuring the criteria weights, the CRITIC method was used. Data obtained from the feedback mechanism were then analyzed using three popular MCDM models, namely TOPSIS, VIKOR, and PROMETHEE II. All the three MCDM models were used to evaluate the usability score of five mHealth applications (which include Glucose Buddy, mySugr, Diabetes: M, Blood Glucose Tracker, and OneTouch Reveal) and provide the ranking of those alternatives. The key findings of the proposed research work are summarized below:The relative proximity ($C_{i}$) obtained from TOPSIS method ranges between 0.271650464 and 0.748031419. The mHealth applications with the maximum ($C_{i}$) value is considered as the best alternative. Here, the mySugr application is considered as the best mHealth application among the rest.The aggregating index ($Q_{a}$) calculated from VIKOR method shows that the alternative having the least ($Q_{a}$) value is the best among all. The ($Q_{a}$) value of the five alternatives taken in our study ranges between 0 and 0.898883249. It was observed that the mySugr application has the least ($Q_{a}$) value among all; thus, it is the best mHealth application as far as usability is concerned.PROMETHEE II is used for calculating the net outranking flow (*φ*(*i*)) of T2DM applications. The (*φ*(*i*)) value obtained in our study ranges between −0.26391893 and 0.344439094. A higher (*φ*(*i*)) value indicates best alternative. From result analysis, it was observed that, in PROMETHEE II, the mySugr application is the best mHealth application with the maximum (*φ*(*i*)) value.A comparison study was carried out among the three MCDM models to check the consistency of the result. It was observed that all the three models show almost the same ranking for the five alternatives, with the mySugr application as the best and the Blood Glucose Tracker as the least preferred application among the users.

This study proposes a very useful methodology for evaluating the usability score of mHealth applications and support decision making in the selection of the best mHealth applications focusing on T2DM patients. This research work is beneficial to patients for their day-to-day health monitoring and recording blood sugar levels, which can also help medical practitioners for further analysis.

Limitations: One of the limitations of the research study is the sample size of the population, which needs to be increased for better efficacy in the decision-making process. This is because the research work was conducted on rare chronic diseases, i.e., T2DM patients, whose population is less in a smaller region of the country, like Jharkhand. Another limitation of the research work is the number of alternatives, because with the increase in the number of alternatives, the MCDM methods may produce varied results.

Future Scope: The future scope of the research work is to develop a hybrid or novel model considering the popular MCDM methods that can provide better result in terms of accuracy. Special attention should be given on fuzzy-based MCDM approaches, which improve expert judgement by removing vagueness and human error. Moreover, feedback will be collected from all groups of users (especially medical experts/doctors/practitioners, etc.) to improve the effectiveness of the decision-making process.

## Figures and Tables

**Figure 1 healthcare-10-00004-f001:**
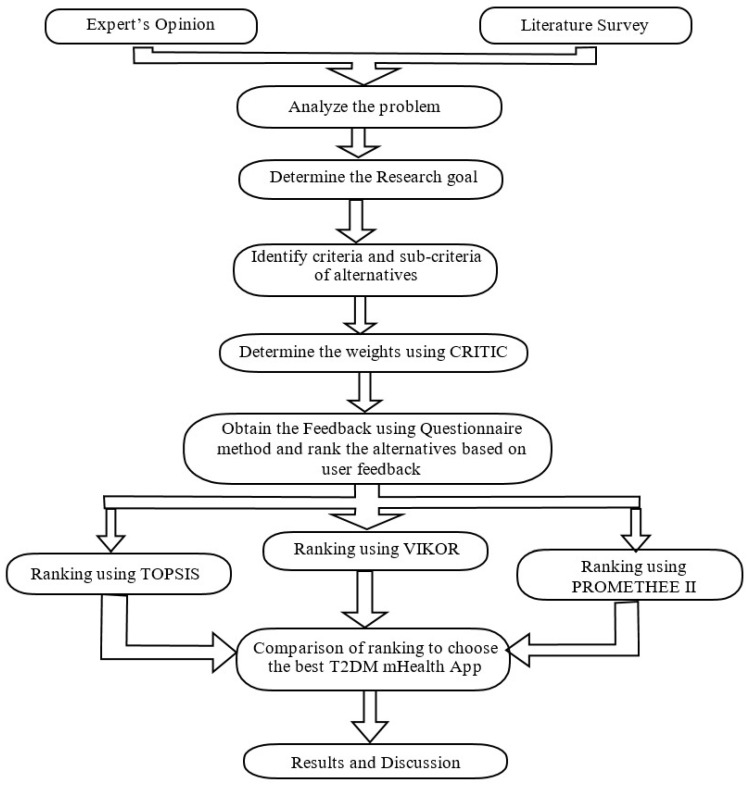
Flowchart model representing the ranking of mHealth application based on usability aspect.

**Figure 2 healthcare-10-00004-f002:**
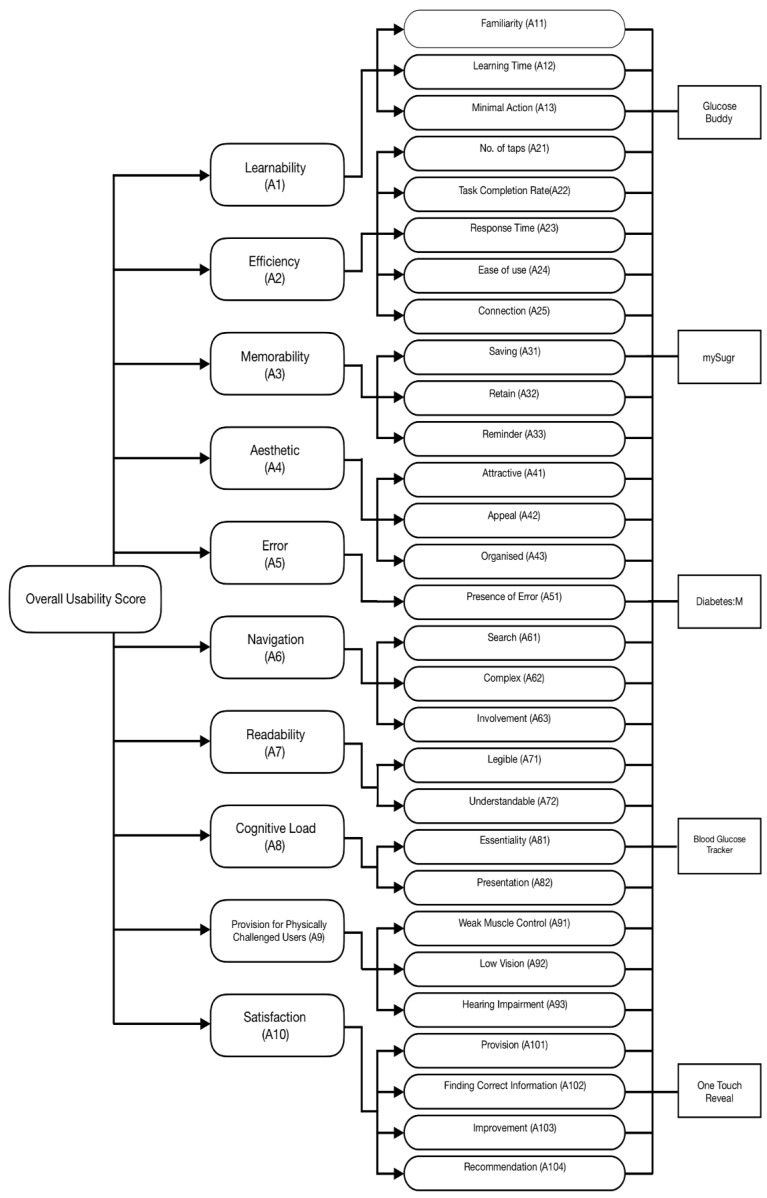
Identifying attributes and sub-attributes for measuring usability score for T2DM mHealth application.

**Figure 3 healthcare-10-00004-f003:**
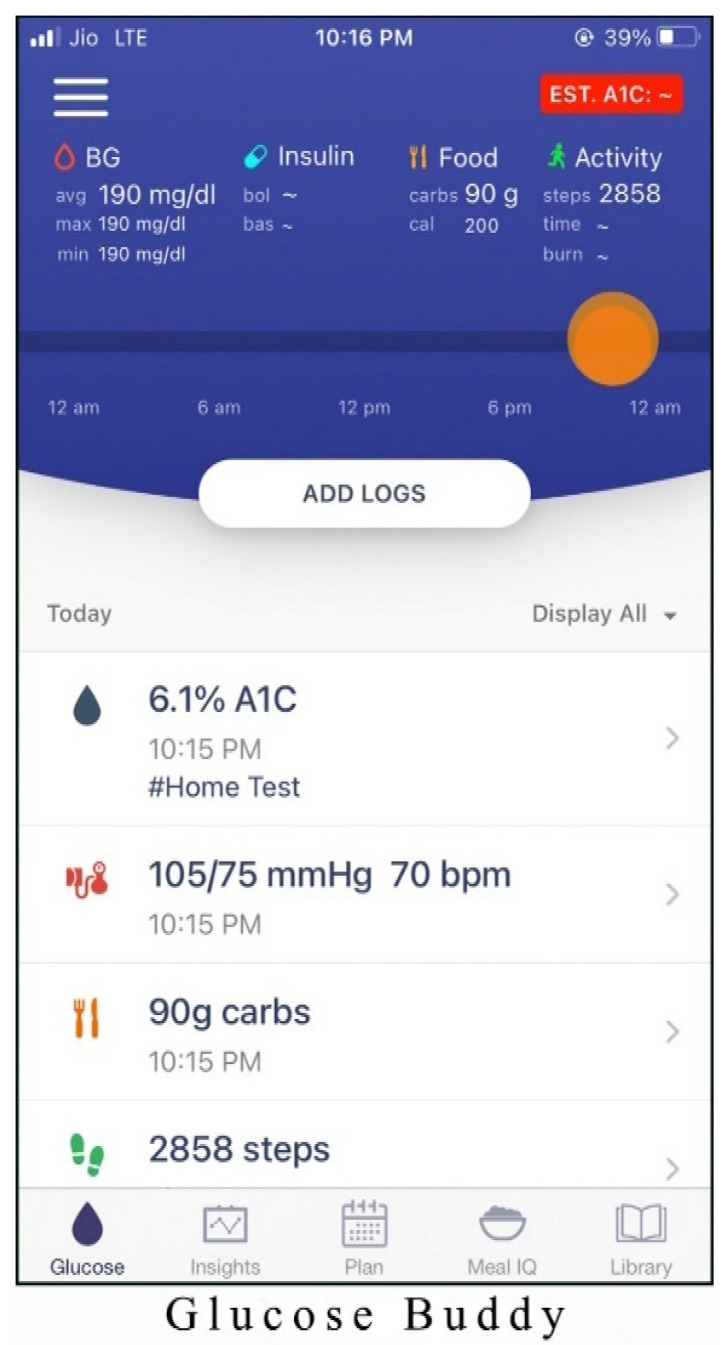
Homepage of Glucose Buddy application.

**Figure 4 healthcare-10-00004-f004:**
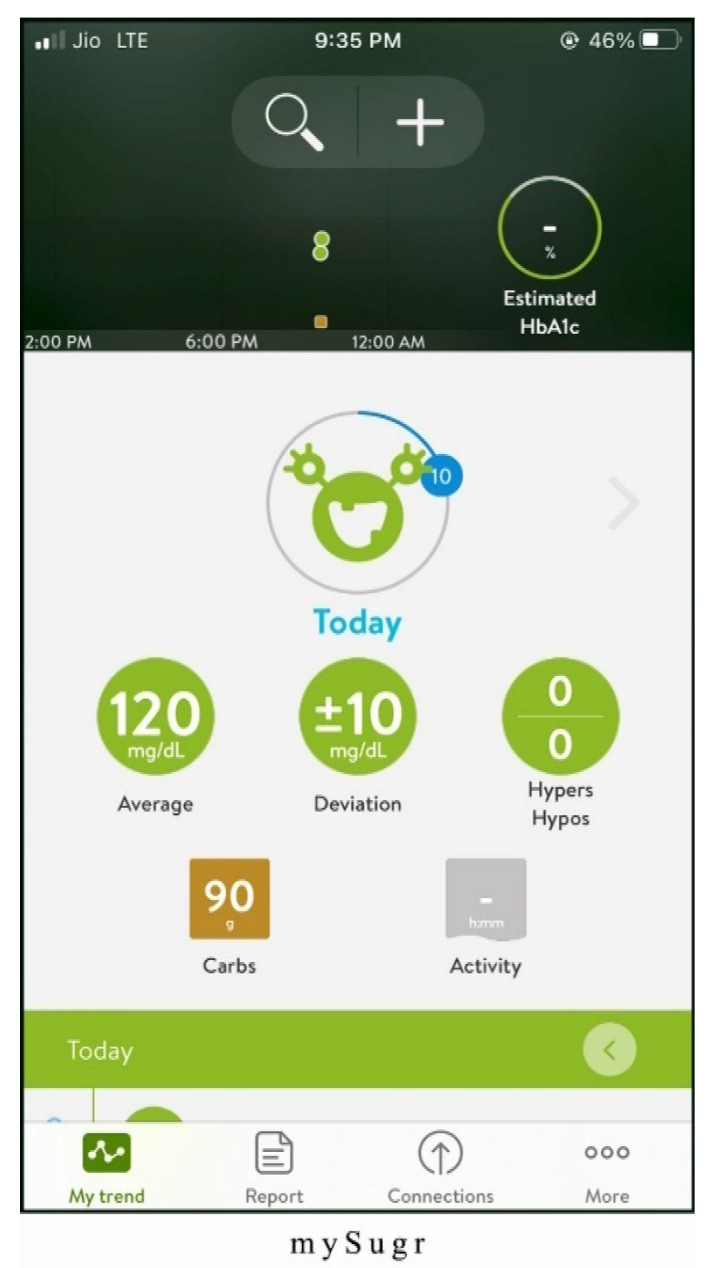
Homepage of mySugr application.

**Figure 5 healthcare-10-00004-f005:**
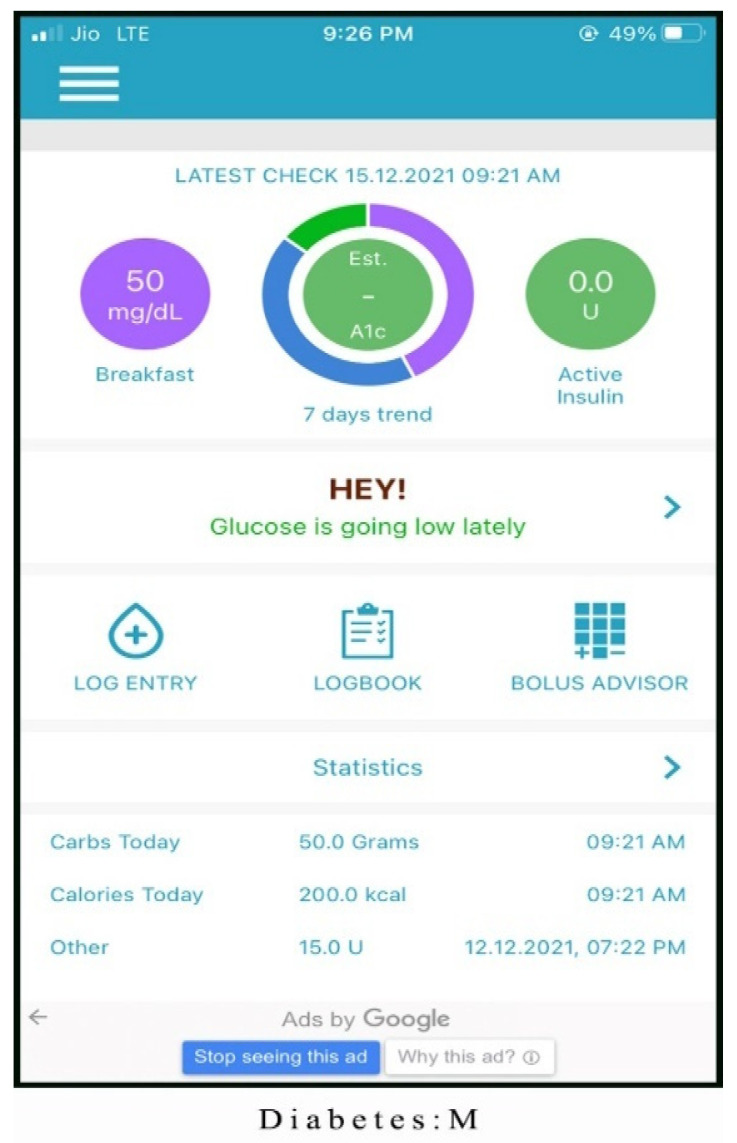
Homepage of Diabetes: M application.

**Figure 6 healthcare-10-00004-f006:**
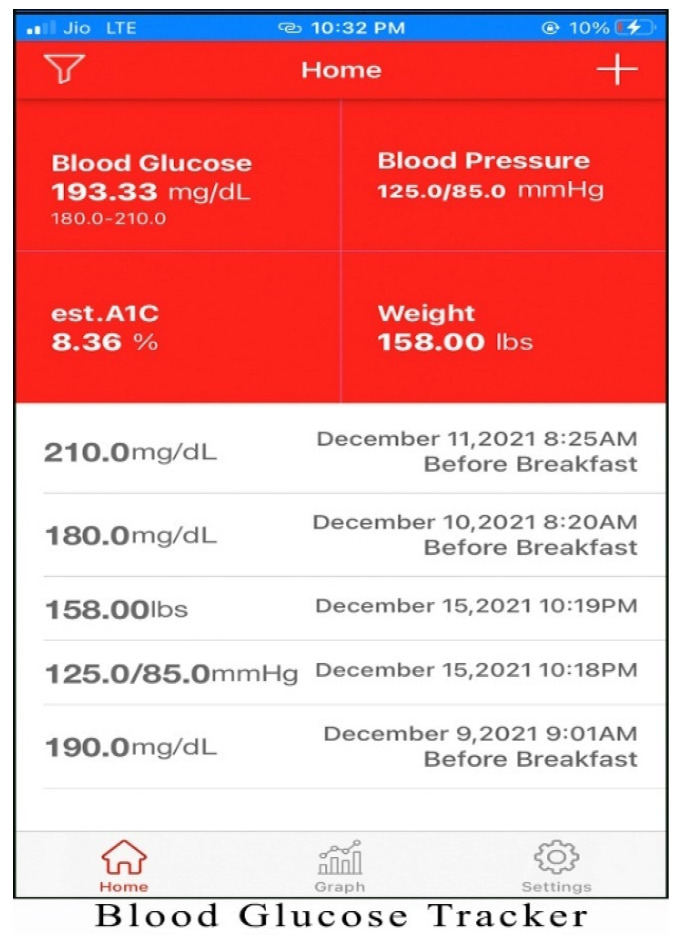
Homepage of Blood Glucose Tracker application.

**Figure 7 healthcare-10-00004-f007:**
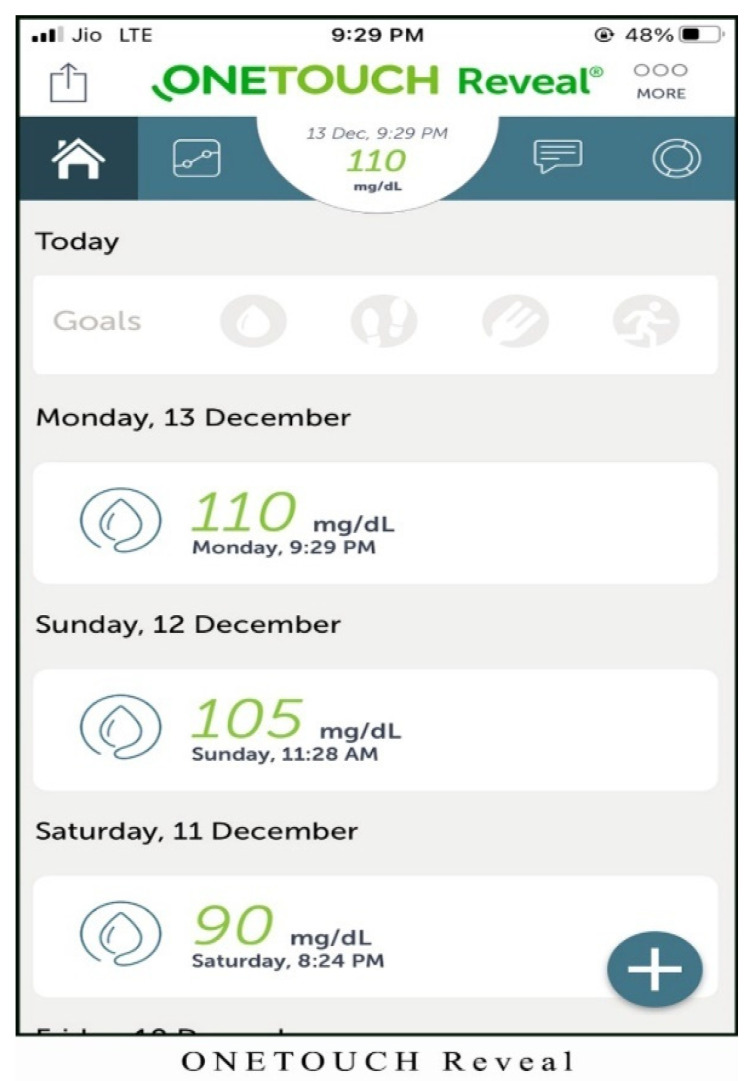
Homepage of OneTouch Reveal application.

**Figure 8 healthcare-10-00004-f008:**
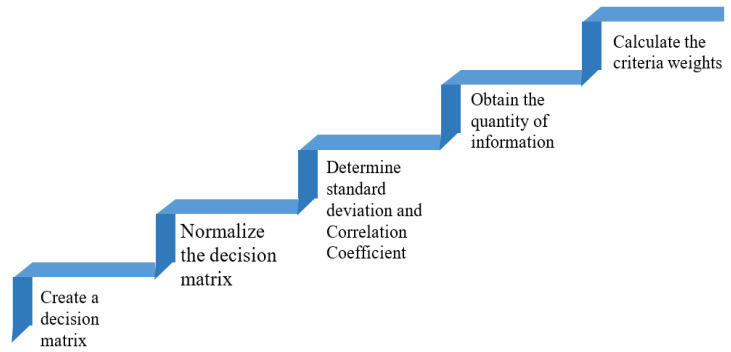
Steps involved in CRITIC method.

**Figure 9 healthcare-10-00004-f009:**
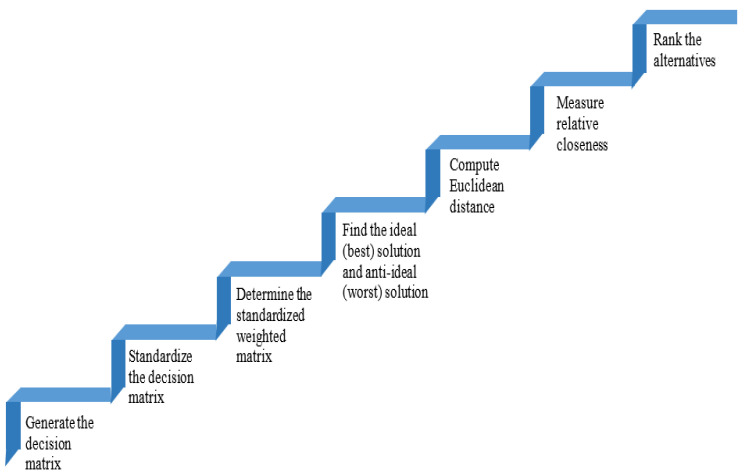
Steps for the technique for order of preference by similarity to ideal solution (TOPSIS) method.

**Figure 10 healthcare-10-00004-f010:**
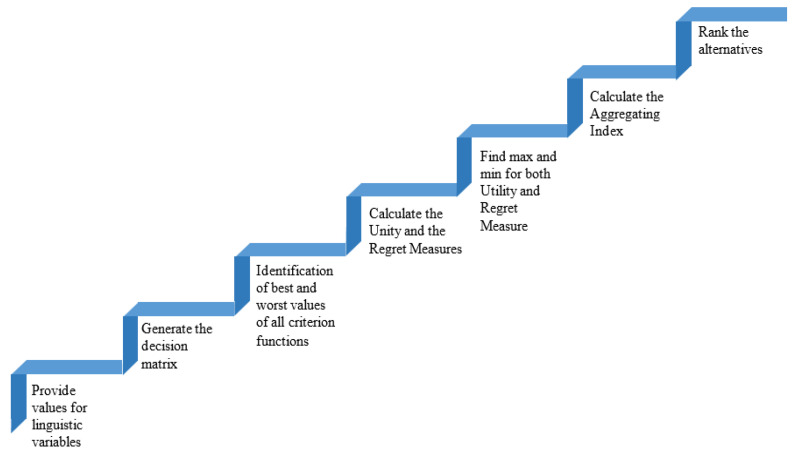
Steps involved in VIKOR method.

**Figure 11 healthcare-10-00004-f011:**
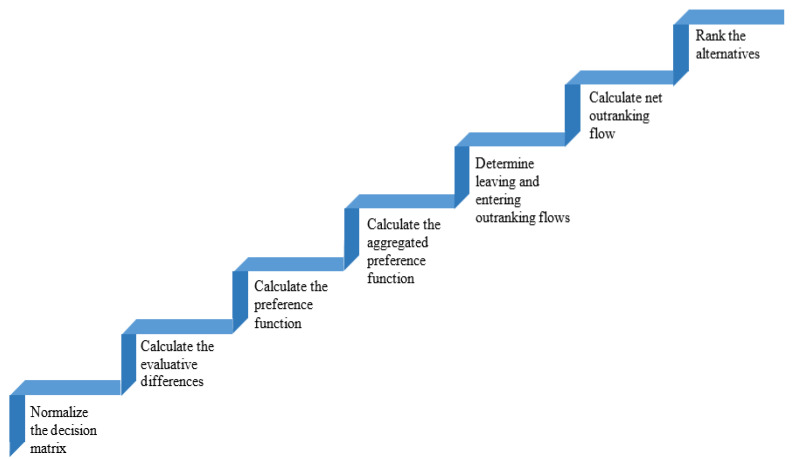
Steps for PROMETHEE II (Preference Ranking Organization Methods for Enrichment Evaluations).

**Figure 12 healthcare-10-00004-f012:**
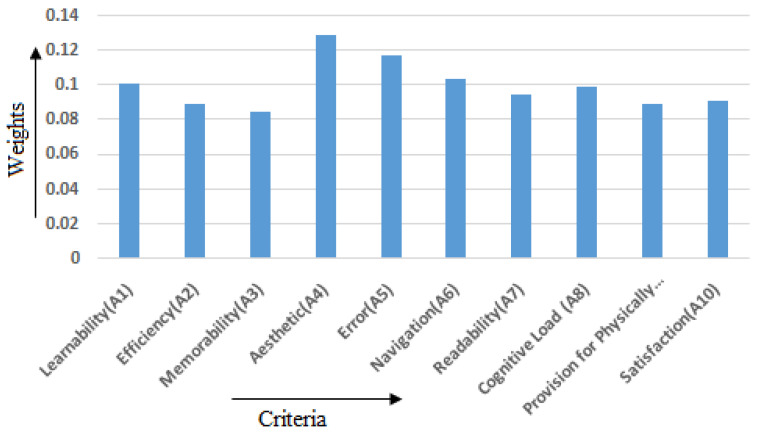
Objective weights of each attribute.

**Figure 13 healthcare-10-00004-f013:**
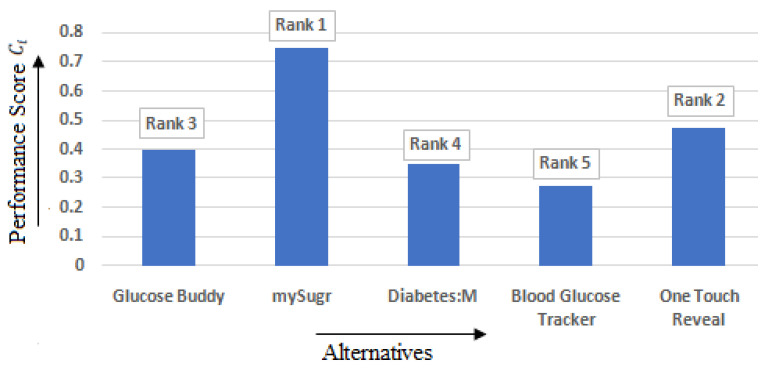
Rank obtained by TOPSIS method based on the performance score.

**Figure 14 healthcare-10-00004-f014:**
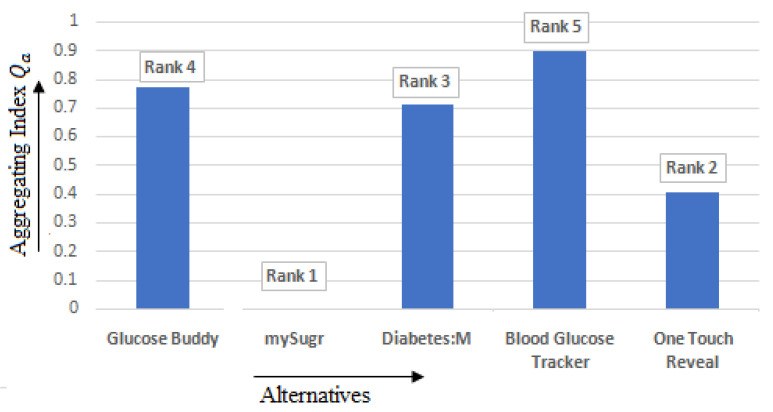
Rank obtained by VIKOR based on the aggregating index.

**Figure 15 healthcare-10-00004-f015:**
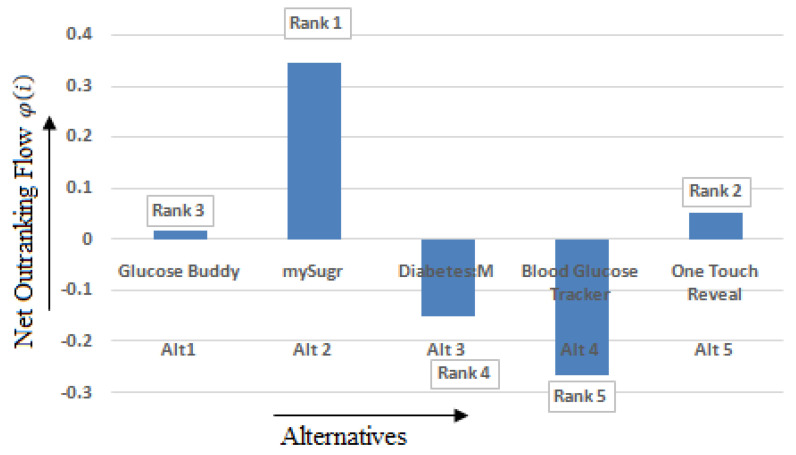
Rank obtained by PROMETHEE II based on the net outranking flow.

**Figure 16 healthcare-10-00004-f016:**
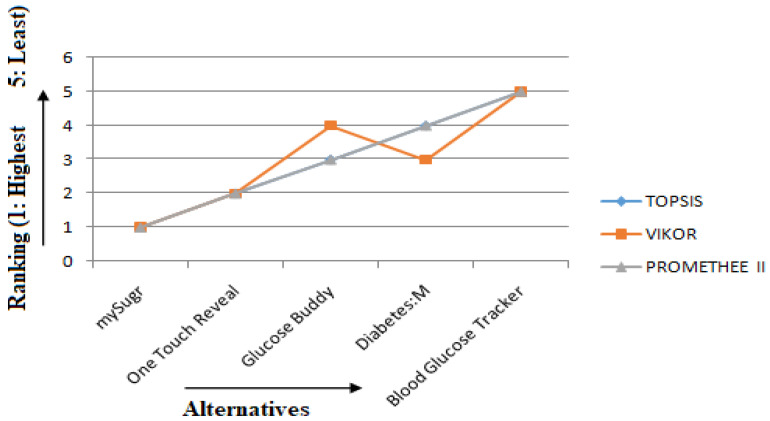
Comparison of the ranks obtained by TOPSIS, VIKOR, and PROMETHEE II.

**Table 1 healthcare-10-00004-t001:** List of attributes (criteria) and sub-attributes (sub-criteria) for identifying the usability score of T2DM mHealth applications.

Attributes (Criteria)	Definition	Sub-Attributes (Sub-Criteria)
Learnability (A1)	The user’s ability to perceive and become familiar with the features and functions of T2DM mHealth applications by applying a minimal amount of effort is referred to as learnability. The higher score means that the T2DM mHealth applications are much more self-descriptive, whereas a low score suggests that the T2DM mHealth applications use terminologies that the user may not be familiar with.	Familiarity (A11)
		Learning time (A21)
		Minimal action (A23)
Efficiency (A2)	The ease of work is measured by efficiency. It denotes how quickly and cheaply users can finish the task given with limited resources.	Number of taps (A21)
		Task completion time (A22)
		Response time (A23)
		Ease-of use (A24)
		Connection (A25)
Memorability (A3)	Memorability refers to how quickly users can re-acquaint themselves with a design after being away from it for a while.	Saving (A31)
		Retain (A32)
		Reminder (A33)
Aesthetic (A4)	For evaluating and analyzing the visual effect of T2DM mHealth applications, aesthetic is a key usability attribute. It helps to determine the users’ interest in the corresponding T2DM mHealth applications, both functionally and non-functionally.	Attractive (A41)
		Appeal (A42)
		Organized (A43)
Error (A5)	During the evaluation of the T2DM mHealth applications, this relates to the frequency of errors made, the seriousness or severity of the errors, and the measures for recovery.	Presence of Error (A51)
Navigation (A6)	Controllability is an important usability attribute that evaluates a T2DM mHealth navigational capability. The score represents how easy it is for the user to navigate through the T2DM mHealth applications and execute the needed task.	Search (A61)
		Complex (A62)
		Involvement (A63)
Readability (A7)	The mHealth applications’ content must be readable. Legibility and understandability are two aspects of readability. Color combinations, word style (italic, bold, etc.), font size, and typeface should all be legible in mHealth applications. In terms of word choice and phrase length, mHealth applications should be understandable.	Legible (A71)
		Understandable (A72)
Cognitive Load (A8)	Cognitive load refers to the number of working memory resources (such as thinking, reasoning, and remembering) required to operate mHealth applications. The mHealth applications’ cognitive load should be reduced. Removing non-essential content, breaking the content into smaller chunks, displaying the information both visually and verbally, and so on are some techniques to reduce cognitive load.	Essentiality (A81)
		Presentation (A82)
Provision for Physically Challenged Users (A9)	A group of users using certain mHealth applications may have physical disabilities such as hearing impairment, movement disabilities, or visual problems, among other things. The application’s user interface should be built or structured in such a way that it can manage various types of user groups as well.	Weak Muscle Control (A91)
		Low Vision (A92)
		Hearing Impairment (A93)
Satisfaction (A10)	This attribute measures the amount of satisfaction with mHealth applications. It refers to the user’s comfort, likeability, and pleasure.	Provision (A101)
		Finding Correct Information (A102)
		Improvement (A103)
		Recommendation (A104)

**Table 2 healthcare-10-00004-t002:** Sub-attributes (sub-criteria) identified for evaluating usability of T2DM mHealth applications.

Sub-Attributes (Sub-Criteria)	Definition
Familiarity (A11)	Measures how quickly the user can easily become familiar with the app.
Learning time (A21)	Measures the average amount of time users spend in learning various app functions.
Minimal action (A23)	It states that minimal action should be required to record/update the blood glucose and other values.
Number of taps (A21)	Measures the number of tapping times needed for searching particular information in the app. Less tapping states that the app is efficient.
Task completion time (A22)	Measures how long it takes people to accomplish a task and compares it to how long it takes an expert to complete the same task. It also specifies that the tasks such as weight management, measure of step count, and tracking of carb intakes, blood glucose values, and HbA1c are completed at a good pace.
Response time (A23)	Measures the response time for recording/charting the glucose values over time.
Ease-of use (A24)	Measures how easily the data, such as blood glucose values, HbA1c, and carbs, are entered/recorded and graphically represented.
Connection (A25)	Measures the efficiency through which the app can be connected to social media and electronic medical record systems.
Saving (A31)	Measures the ease with which the blood sugar values, medication log, and carb intakes are saved for future reference.
Retain (A32)	Measures how comfortably the blood glucose values, medication names, carb intakes, and other values are retained for a long time.
Reminder (A33)	Measures how efficiently the reminder/alert function is set.
Attractive (A41)	Measures the extent at which the layout of the app is found attractive by its users.
Appeal (A42)	Measures as to what extent the app’s design visually appeals to the users.
Organized (A43)	Measures how meaningfully the app features are organized.
Presence of Error (A51)	It states whether the app contains errors and measures the frequency of errors that result from users and compares it with the target value.
Search (A61)	Measure the time taken by the app to search for food databases to log in meals.
Complex (A62)	Measures the complexity of the navigation of the app features.
Involvement (A63)	It measures the user engagement and intervention in the app for finding information.
Legible (A71)	It states that the app properties and features should be legible in all aspects.
Understandable (A72)	Measures the extent to which the basic characteristics and features of the app are clearly described and can be easily understood by users.
Essentiality (A81)	It specifies that only the essential components are present in the app.
Presentation (A82)	It states that the information should be presented both visually and verbally.
Weak Muscle Control (A91)	Measures the efficiency through which the app can be handled by users with weak muscle control.
Low Vision (A92)	This specifies that there should be a provision of increasing the font size for helping users with low vision.
Hearing Impairment (A93)	This assures that valuable information such as app education should be available in the text along with the audio to assist the user with hearing impairment.
Provision (A101)	Measures as to what extent the app provides resources/tips to the users.
Finding Correct Information (A102)	This attribute specifies that the app should be helpful in finding the correct information.
Improvement (A103)	Measures the extent to which the app is capable of improving lifestyle.
Recommendation (A104)	It measures the action of the users in recommending the app to others.

**Table 3 healthcare-10-00004-t003:** Demographics of the participants (n = 30).

Variable		n
Age	<65	20
	≥65	10
Gender	Male	15
	Female	15
Use of Smart phone	Frequently (4 to 7 days in a week)	26
	Occasionally (1 to 3 days in a week)	4
	Rarely (<1 day in a week)	0
Education	Elementary School	1
	Middle School	2
	High School	7
	Graduate	14
	Post Graduate	6

**Table 4 healthcare-10-00004-t004:** Usability score of each attribute.

Alternatives ID	Alternatives Name	A1	A2	A3	A4	A5	A6	A7	A8	A9	A10
Alt 1	Glucose Buddy	119	104	98	101	82	70	130	81	117	119
Alt 2	mySugr	128	115	113	111	63	72	113	64	111	116
Alt 3	Diabetes: M	120	102	98	121	91	74	107	74	111	115
Alt 4	Blood Glucose Tracker	121	101	99	116	94	73	111	75	109	104
Alt 5	OneTouch Reveal	121	99	92	120	82	69	113	68	111	115

**Table 5 healthcare-10-00004-t005:** Weight of each attribute.

Attributes	Learnability (A1)	Efficiency (A2)	Memorability (A3)	Aesthetic (A4)	Error (A5)	Navigation (A6)	Readability (A7)	Cognitive Load (A8)	Provision for Physically Challenged Users (A9)	Satisfaction (A10)
Weights	0.101316801	0.089114537	0.084815071	0.129194116	0.117180195	0.104248405	0.094418961	0.099547741	0.089309841	0.090854332

**Table 6 healthcare-10-00004-t006:** Rank based on $C_{i\;}$.

Alternatives ID	Item (Alternatives)	$C_{i}\;(\mathrm{Performance}\;\mathrm{Score}\;\mathrm{or}\;\mathrm{Relative}\;\mathrm{Closeness})$	Rank
Alt 1	Glucose Buddy	0.397278402	3
Alt 2	mySugr	0.748031419	1
Alt 3	Diabetes: M	0.350935988	4
Alt 4	Blood Glucose Tracker	0.271650464	5
Alt 5	OneTouch Reveal	0.47378464	2

**Table 7 healthcare-10-00004-t007:** Rank obtained based on the values of $Q_{a}$ (aggregating index).

Alternatives ID	Item (Alternatives)	$Q_{a}\;(\mathrm{Aggregating}\;\mathrm{Index}\;\mathrm{or}\;\mathrm{the}\;\mathrm{Final}\;\mathrm{Negotiation}\;\mathrm{Solution})$	Rank
Alt 1	Glucose Buddy	0.769670172	4
Alt 2	mySugr	0	1
Alt 3	Diabetes: M	0.709485465	3
Alt 4	Blood Glucose Tracker	0.898883249	5
Alt 5	OneTouch Reveal	0.40239348	2

**Table 8 healthcare-10-00004-t008:** Rank obtained based on the values of φ(i) (net outranking flow).

Alternatives ID	Item (Alternatives)	φ(i) (Net Outranking Flow)	Rank
Alt 1	Glucose Buddy	0.01632707	3
Alt 2	mySugr	0.344439094	1
Alt 3	Diabetes: M	−0.14960478	4
Alt 4	Blood Glucose Tracker	−0.26391893	5
Alt 5	OneTouch Reveal	0.052757542	2

**Table 9 healthcare-10-00004-t009:** Comparison of the alternatives based on the usability rank.

(Alternatives) Item	TOPSIS	VIKOR	PROMETHEE II
Alt 2 (mySugr)	Rank 1	Rank 1	Rank 1
Alt 5 (OneTouch Reveal)	Rank 2	Rank 2	Rank 2
Alt 1 (Glucose Buddy)	Rank 3	Rank 4	Rank 3
Alt 3 (Diabetes: M)	Rank 4	Rank 3	Rank 4
Alt 4 (Blood Glucose Tracker)	Rank 5	Rank 5	Rank 5

## Data Availability

Data sharing not applicable to this article as no datasets were generated or analyzed during the current study.

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
