# Peer review of "Evaluating the Usability of mHealth Applications on Type 2 Diabetes Mellitus Using Various MCDM Methods"

_healthcare, 2021, doi:10.3390/healthcare10010004_

Round 1

Reviewer 1 Report

The authors have conducted a research on the use of various MCDM methods to evaluate the usability of mHealth applications on type-2 diabetes mellitus. My main concern is the lack of justification for use of several MCDM methods. My comments are as follows.

  • Section 3.1.1, present the attributes in a Table, not in the text. Or the authors may just delete the sub-attributes from the text as they have been presented in Figure 1.
  • Line 324, do not copy and paste the equations. Also, each equation should be numbered.
  • Why did the authors need to adopt several methods and compare the findings? This question arises because no novel/ hybrid technique is developed in this manuscript that requires comparative analysis. The authors have to well justify this issue. As mentioned by the authors in lines 617-619 and other parts of the manuscript, the authors wanted to check the consistency of the results which is not necessary from the reviewer’s point of view. Since all used MCDM methods are well established, the result would be similar, and of course not completely the same.
  • Moreover, what is the reason for using these MCDM methods? There are many other methods. For example, why did not the authors use ANP?
  • Line 621, the authors should show the flowchart of their research methodology in the Methodology section.
  • Limitations of the research and future work should be highlighted in the conclusion.

Author Response

Reviewer#1

The authors have conducted a research on the use of various MCDM methods to evaluate the usability of mHealth applications on type-2 diabetes mellitus. My main concern is the lack of justification for use of several MCDM methods. My comments are as follows.

  1. Section 3.1.1, present the attributes in a Table, not in the text. Or the authors may just delete the sub-attributes from the text as they have been presented in Figure 1.

Response: The attributes along with definition are presented in the Table 1 and the corresponding sub-attributes are defined in Table 2.

  1. Line 324, do not copy and paste the equations. Also, each equation should be numbered.

Response: The equation has been modified and reflected in the manuscript. Numbering of the equations is also completed.

  1. Why did the authors need to adopt several methods and compare the findings? This question arises because no novel/ hybrid technique is developed in this manuscript that requires comparative analysis. The authors have to well justify this issue. As mentioned by the authors in lines 617-619 and other parts of the manuscript, the authors wanted to check the consistency of the results which is not necessary from the reviewer’s point of view. Since all used MCDM methods are well established, the result would be similar, and of course not completely the same.

Response: The reviewer has raised a valid question regarding the adoption of several methods and comparing the result. Authors in this research work have adopted several methods based on one of the research article by Salabun et al. (SaÅ‚abun W, WÄ…tróbski J, Shekhovtsov A. Are MCDA Methods Benchmarkable? A Comparative Study of TOPSIS, VIKOR, COPRAS, and PROMETHEE II Methods. Symmetry. 2020; 12(9):1549. https://doi.org/10.3390/sym12091549) where a comparative analysis has been carried among several Multi-Criteria Decision Analysis (MCDA) methods, to benchmark some selected methods namely TOPSIS, VIKOR, PROMETHEE II and COPRAS. The analysis was performed based on various weighing methods (such as entropy and standard deviation) and with various techniques of normalization of MCDA model input data. It has been observed from the result that despite several MCDA methods available, no single method is perfect and cannot be considered suitable for any decision making problem. Because not only the methods itself, but also the combination of normalization and other parameters produce different results. Results also vary with the increase in the number of alternatives. The research work conducted aims in bringing out the similarity of rankings obtained using the four selected methods. It has been observed from the study that TOPSIS, VIKOR PROMETHEE II and COPRAS are considered as the best and popular MCDA methods and are suitable in diverse domain like sustainability assessment, healthcare management, supplier selection, environment management and many more.

The results obtained from the study also signify that the ranking obtained varies depending on the weights and normalization factor. It has been observed that ranking obtained using VIKOR is less correlated as compared to ranking obtained by the other three methods. Hence, authors in this manuscript decided to check the consistency of the results obtained by the three MCDM methods.

This has been reflected under Discussion section 4.4.5 of the manuscript.

Implementation of hybrid/ novel MCDM technique is kept in future scope of the study.

  1. Moreover, what is the reason for using these MCDM methods? There are many other methods. For example, why did not the authors use ANP?

Response: From the research work conducted by Salabun et al. (SaÅ‚abun W, WÄ…tróbski J, Shekhovtsov A. Are MCDA Methods Benchmarkable? A Comparative Study of TOPSIS, VIKOR, COPRAS, and PROMETHEE II Methods. Symmetry. 2020; 12(9):1549. https://doi.org/10.3390/sym12091549) it has also been observed that TOPSIS, VIKOR and PROMETEE II are most popular MCDM methods and are efficient for taking decision in the case where there are fewer alternatives.

ANP (Analytic Network Process) is also an MCDA method where the decision problem is structured in network form. It works better where there is interdependence among the hierarchy elements (i.e. criteria, sub-criteria and alternatives). It may not give efficient result if the criteria and alternatives are independent of each other.

  1. Line 621, the authors should show the flowchart of their research methodology in the Methodology section.

Response: The flowchart of the research methodology of the proposed work is reflected in section 3 (Figure 1) under ‘Proposed Methodology’.

  1. Limitations of the research and future work should be highlighted in the conclusion.

Response:

Limitation: One of the limitations of the research study is the sample size of the population, which needs to be increased for better efficacy in decision making process. This is because the research work has been conducted on rare chronic disease i.e. T2DM, whose population is less in a smaller region of the country, like Jharkhand. Another limitation of the research work is the number of alternatives. Because with the increase in the number of alternatives, the MCDM methods may produce varied results.

Future Work: The future scope of the research work is to develop a hybrid or novel model considering the popular MCDM methods which can provide better result in terms of accuracy. Special attention should be given on fuzzy based MCDM approaches which improve expert judgement by removing vagueness and human error. Moreover, feedback will be collected from all groups of users (specially medical experts/ doctors/ practioners etc.) to improve the effectiveness of the decision making process.  

Limitation and future work has been highlighted in Conclusion section of the manuscript.

Reviewer 2 Report

Usability studies are important to avaluate applications with the participation of end-users and this study contributes to the benchmarking of 5 mHealth applications for Diabites. Auhors have explained their methodology and aims and presented the results of the evaluation with the help of 30 people.

Authors need to explain in the conclusions how the improvement of usability and efficiency of mHealth applications related to expected health ourcomes. Decision making in the selection of best mHealth applications cannot be based only on usability and user experience issues, but also in experts (medical doctors) opinion.
It was not very clear on which criteria those five apps were selected for evaluation. Being part of the same platform does not make things more clear. There are also a lot of alternatives which could be evaluated and somehow be used as a reference. Otherwise it is dofficult to justify the positioning of the propsoed apps.
Finally, readers may need to have a picture of the named apps (screenshots), and a better description of what kind of serivces they offer.

Author Response

Reviewer#2

Usability studies are important to evaluate applications with the participation of end-users and this study contributes to the benchmarking of 5 mHealth applications for Diabetes. Authors have explained their methodology and aims and presented the results of the evaluation with the help of 30 people.

Response: Thank you for your valuable suggestion.

  1. Authors need to explain in the conclusions how the improvement of usability and efficiency of mHealth applications related to expected health outcomes.

Response: As we all know the present doctor-to-patient ratio in our country, it is very much essential to digitalize the health services. The only way to do so is through mHealth applications, which become the medium for delivering health services in real time and also facilitate consultation from remote location. However, all these facilities require efficient mHealth application with good user interface design for improving task effectiveness and user satisfaction. Thus usability evaluation will surely help in identifying the best of mHealth applications which in turn improve the health outcomes by improving patient experience of care and facilities provided on time, saving cost and time while visiting doctor’s chamber, facilitating real time health monitoring and many more.

This has been reflected in Conclusion part.

  1. Decision making in the selection of best mHealth applications cannot be based only on usability and user experience issues, but also in experts (medical doctors) opinion.

Response: Authors agree with reviewer’s suggestion. However, for the time constraint the study is limited to only the users/ patients. Feedback from medical doctors are kept in future scope of the study. However, doctors/ medical practioners suggestions has been taken while preparing the task list that has been carried out by the users.

  1. It was not very clear on which criteria those five apps were selected for evaluation. Being part of the same platform does not make things more clear.

Response: The five mHealth applications that are selected in this study, all are used for monitoring patients suffering from Type 2 Diabetes Mellitus disease. These mHealth applications are considered as best apps as per Healthline report 2020 [https://www.healthline.com/health/diabetes/top-iphone-android-apps]. Thus, authors in this work want to evaluate the usability aspect of the applications to measure the efficiency and satisfaction level of the users while interacting with the interface.

Moreover, the authors have taken suggestions from medical experts before considering these mHealth applications.

  1. There are also a lot of alternatives which could be evaluated and somehow be used as a reference. Otherwise it is difficult to justify the positioning of the proposed apps.

Response: Less number of alternatives chosen for the study is one of the limitations of the research work. This point will be addressed in future scope of the work.

  1. Finally, readers may need to have a picture of the named apps (screenshots), and a better description of what kind of services they offer.

Response: Alternatives chosen for the study are one of the best mHealth applications for monitoring diabetes patients. The services offered by these applications are mentioned below:

mySugr: The best features of mySugr application are as follows:

  • Personalized logging screen which can record the value from Bluetooth-enabled blood glucose meter and analyze the pattern to brief the blood glucose levels.
  • Smart search facility for recording meals and activities which helps in controlling diabetes.
  • Ability to provide highest quality security as per General Data Protection Regulation (GDPR).

Glucose Buddy: services offered are:

  • Simple and hassle free solution for controlling diabetes with real time blood sugar measurement.
  • Provides professional support and advice.

Diabetes:M : this mHealth application provides everything needed for effective health management by offering the following services:

  • Provides detailed information of user.
  • Effective diabetes control remotely.
  • Present the report in statistical format (like chart, bar) which produce better understanding among the users.
  • Feature of recognizing the pattern and look for any predefined recurring problems along with the reason for occurrence.
  • Adds an insulin bolus calculator for calculating insulin based on nutritional information.

Blood Glucose Tracker: the services offered by this application are as follows:

  • Tracking blood glucose at every level (like breakfast, lunch, and dinner) throughout the day, thereby helping patients to control blood sugar efficiently.
  • Moreover, it can also record blood pressure, weight, HbA1c etc.

One Touch Reveal: the unique features of this mHealth application are:

  • It provides unique colour coding technology to organize blood sugar result which can be easily understood by naïve users.
  • It automatically notifies repeated highs or lows so that proper action can be taken.
  • This alternative also sets the goal for recording steps walked daily, carbs and activity as well.
  • It set reminders for undergoing blood sugar test as well as for taking insulin.

This has been reflected in section 3.2 along with the snapshots of the applications provided in Figure 3, 4, 5,  6, and 7.

Reviewer 3 Report

The article is within the scope of the journal.
The subject of the article is interesting for the area of knowledge.

It is well written and structured. Reading is fluent.

The results obtained are an advance for the problem raised. The experiment is well designed and the interpretation of the results shows interesting data.

However, the article needs to be improved:
a) It would be necessary to introduce a discussion section in which the work presented is compared with other similar ones and the progress regarding the current state of the issue as well as the limitations is shown. Recent bibliography should be used for this.
b) In the conclusion section, the presentation of the scientific contribution made should be improved. Likewise, it should be indicated what the future lines of work would be.

Author Response

The article is within the scope of the journal.
The subject of the article is interesting for the area of knowledge.

Response: Thank you for reviewer comments

  1. It is well written and structured. Reading is fluent.

Response: Thank you for reviewer comments

  1. The results obtained are an advance for the problem raised. The experiment is well designed and the interpretation of the results shows interesting data.

Response: Thank you for the reviewer comments

However, the article needs to be improved:

a). It would be necessary to introduce a discussion section in which the work presented is compared with other similar ones and the progress regarding the current state of the issue as well as the limitations is shown. Recent bibliography should be used for this.

Response: Discussion and comparison of the proposed work has been reflected in section 4.4.5 under the Result analysis of the manuscript.

Limitations of the study are also mentioned in the Conclusion section.

Recent bibliography is included in Related Work as well as Reference section of the manuscript numbered from 24 to 32.

b) In the conclusion section, the presentation of the scientific contribution made should be improved. Likewise, it should be indicated what the future lines of work would be.

Response: The scientific contribution made in the paper are improved and presented below. It is also reflected in the Conclusion section.

Scientific Contribution:

The key findings of the proposed research work are summarized below:

  • The relative-proximity ( ) obtained from TOPSIS method ranges between 0.271650464 and 0.748031419. The mHealth applications having the maximum ( ) value is considered as the best alternative. Here, mySugr application is considered as the best mHealth application among the rest.
  • The Aggregating Index ( ) calculated from VIKOR method shows that the alternative having least ( ) value is the best among all. The ( ) value of the five alternatives taken in our study ranges between 0 and 0.898883249. It has been observed that mySugr application is having least ( ) value among all, thus it is the best mHealth application as per usability is concerned.
  • PROMETHEE II is used for calculating net outranking flow (φ(i)), of T2DM applications. The (φ(i)) value obtained in our study ranges between -0.26391893 and 0.344439094. Higher (φ(i)) value indicates best alternative. From result analysis, it has been observed that in PROMETHEE II also my-sugr application is the best mHealth application with maximum (φ(i)) value.
  • A comparison study has been carried out among the three MCDM models to check the consistency of the result. It has been observed that all the three models have shown almost the same ranking for the five alternatives with mySugr application as the best and Blood Glucose Tracker as the least preferred application among the users.

The future scope of the research work is to develop a hybrid or novel model considering the popular MCDM methods which can provide better result in terms of accuracy. Special attention should be given on fuzzy-based MCDM approaches which improve expert judgment by removing vagueness and human error. Moreover, feedback will be collected from all groups of users (especially medical experts/ doctors/ practitioners etc.) to improve the effectiveness of the decision-making process.  

This has been reflected in the conclusion section.

Reviewer 4 Report

The topic is potentially interesting. The approach used is technically correct, but not innovative. The sample size is also quite small. Significant improvements are needed in several areas.

Improvement suggestions:

  1. Authors state: “The One of the most common chronic disorders that affect people today is Diabetes Mellitus, and its incidence is quickly increasing around the world”. It would be interesting to know the incidence levels.
  2. Authors define “Usability is a qualitative criterion or attribute that can be defined as simplicity and ease of use….” A clearer definition is needed and based in the literature.
  3. What the authors want to say with “standard questionnaire”. What is the concept of standard?
  4. It is not clear how the 10 usability attributes were defined. More detail about it should be given.
  5. Table 1 is not needed. It only contains obvious information.
  6. The sub-attributes should also be defined. It would be a good idea to present them into a table.
  7. The quality of Figure 1 should also be improved. The same applies to Figure 3, Figure 4, and Figure 5.
  8. It is not clear how the participants were chosen. What is the risk of bias? The number of participants looks relatively low.
  9. The quality of Figure 8, and 9 should also be improved.
  10. The paper should have a discussion section to discuss the relevance of these results and compare it against other published studies.
  11. The conclusions section should also present its limitations. Furthermore, future research directions should be given.
  12. Some references are wrongly formatted. Please for example the first reference. It is not the way to cite a web site.
  13. The number of references should be increased an include other relevant studies like:

https://mhealth.jmir.org/2019/1/e12160/

https://mhealth.jmir.org/2019/4/e11500

https://bmcmedinformdecismak.biomedcentral.com/articles/10.1186/s12911-020-1033-3

https://www.hindawi.com/journals/jdr/2016/1604609/

https://www.sciencedirect.com/science/article/pii/S1532046415002762

https://mhealth.jmir.org/2021/2/e23477/

https://ieeexplore.ieee.org/document/8531158

https://conservancy.umn.edu/handle/11299/206282

https://dergipark.org.tr/en/pub/clinexphealthsci/issue/53248/599548

  1. In particular, the authors should identify the main differences between their study and this last reference. There are several common points.

Author Response

The topic is potentially interesting. The approach used is technically correct, but not innovative. The sample size is also quite small. Significant improvements are needed in several areas.

Improvement suggestions:

1. Authors state: “The One of the most common chronic disorders that affect people today is Diabetes Mellitus, and its incidence is quickly increasing around the world”. It would be interesting to know the incidence levels.

Response: According to World Health Organization (WHO) report 2021 on Diabetes, the following incidence levels are observed which are mentioned below:

  • The number of people with diabetes rose from 108 million in 1980 to 422 million in 2014.
  • In 2014, 8.5% of adults aged 18 years and older had diabetes.
  • In 2019, diabetes was the direct cause of 1.5 million deaths and 48% of all deaths due to diabetes occurred before the age of 70 years.
  • Between 2000 and 2016, there was a 5% increase in premature mortality rates (i.e. before the age of 70) from diabetes. In high-income countries the premature mortality rate due to diabetes decreased from 2000 to 2010 but then increased in 2010-2016. In lower-middle-income countries, the premature mortality rate due to diabetes increased across both periods.
  • Type 2 diabetes was seen only in adults but it is now also occurring increasingly frequently in children.

This has been reflected in Introduction section 1.

2. Authors define “Usability is a qualitative criterion or attribute that can be defined as simplicity and ease of use….” A clearer definition is needed and based in the literature.

Response: Usability is defined as “the extent to which a product can be used by specified users to achieve specified goals with effectiveness, efficiency and satisfaction in a specified context of use” [ISO, 1998]. According to Jakob Nielsen (1993), “Usability is the measure of the quality of the user experience when interacting with something- whether a website, a traditional software application or any other device the user can operate in some way or another.”

This definition is reflected under Proposed Methodology (Section 3)

3. What the authors want to say with “standard questionnaire”. What is the concept of standard?

Response: The questionnaires are prepared by following WAMMI (Website Analysis and MeasureMent Inventory), one of the world-wide standard psychometric scientific analytics service, iteratively developed using Psychometric techniques. It has been scientifically proven and has a reliability data rating of between 0.90 and 0.93. Standard here means that follows already accepted and proven questionnaire for usability evaluation of interface of software application.

4. It is not clear how the 10 usability attributes were defined. More detail about it should be given.

Response: The attributes defined for evaluating the usability of mHealth applications are identified based on numerous literature survey as well as perspective-based User Interface inspection. In this study, perspective is gathered from the user profiles, their technical ability, and most important the medical practioners who facilitate the mHealth applications by rendering different services and support to the patients for regular health monitoring related to T2DM. Various tasks were also been carried out by the inspectors to explore the mHealth applications with focus on the different perspectives.

This part is included in section 3.1.1

5. Table 1 is not needed. It only contains obvious information.

Response: Table 1 is removed as per suggestion.

6. The sub-attributes should also be defined. It would be a good idea to present them into a table.

Response: The sub-attributes are defined in Table 2 in the manuscript.

7. The quality of Figure 1 should also be improved. The same applies to Figure 3, Figure 4, and Figure 5.

Response: The figures are changed and quality is improved and reflected in the manuscript.

8. t is not clear how the participants were chosen. What is the risk of bias? The number of participants looks relatively low.

Response: The participants chosen are from one smaller region of the country (Jharkhand) who is suffering from Type 2 Diabetes Mellitus, having sufficient technical knowledge and fluency in using smart phone.

Less number of participants is one of the limitations of the study which will be taken care in future research work.

9. The quality of Figure 8, and 9 should also be improved.

Response: The figures are changed and quality is improved and reflected in the manuscript.

10. The paper should have a discussion section to discuss the relevance of these results and compare it against other published studies.

Response: Discussion and comparison of the proposed work has been reflected in section 4.4.5 under Result analysis of the manuscript.

11. The conclusions section should also present its limitations. Furthermore, future research directions should be given.

Response:

Limitation: One of the limitations of the research study is the sample size of the population, which needs to be increased for better efficacy in decision making process. This is because the research work has been conducted on rare chronic disease i.e. T2DM, whose population is less in a smaller region of the country, like Jharkhand. Another limitation of the research work is the number of alternatives. Because with the increase in the number of alternatives, the MCDM methods may produce varied results.

Future Work / Research Directions: The future scope of the research work is to develop a hybrid or novel model considering the popular MCDM methods which can provide better result in terms of accuracy. Special attention should be given on fuzzy based MCDM approaches which improve expert judgement by removing vagueness and human error. Moreover, feedback will be collected from all groups of users (specially medical experts/ doctors/ practioners etc.) to improve the effectiveness of the decision making process.  

Limitation and future work has been highlighted in Conclusion section of the manuscript.

12. Some references are wrongly formatted. Please for example the first reference. It is not the way to cite a web site.

Response: References are modified and mentioned correctly in the Reference section of the manuscript.

13. The number of references should be increased an include other relevant studies like:

https://mhealth.jmir.org/2019/1/e12160/

https://mhealth.jmir.org/2019/4/e11500

https://bmcmedinformdecismak.biomedcentral.com/articles/10.1186/s12911-020-1033-3

https://www.hindawi.com/journals/jdr/2016/1604609/

https://www.sciencedirect.com/science/article/pii/S1532046415002762

https://mhealth.jmir.org/2021/2/e23477/

https://ieeexplore.ieee.org/document/8531158

https://conservancy.umn.edu/handle/11299/206282

https://dergipark.org.tr/en/pub/clinexphealthsci/issue/53248/599548

Response: All the above references have been incorporated under section 2 i.e. Review Works. References are also cited in reference section numbered from 24 to 32.

14. In particular, the authors should identify the main differences between their study and this last reference. There are several common points.

Response: The difference between the proposed study and other research work is explained under Result analysis section 4.4.5 ‘Discussion and comparison of the Proposed Methodology’.

Round 2

Reviewer 1 Report

The authors have addressed my comments and the manuscript can be published in Healthcare.

Author Response

The authors have addressed my comments and the manuscript can be published in Healthcare.

Response: Thank you for your suggestions, which improve the quality of the research work. 

Reviewer 3 Report

The paper can be accepted in current form

Author Response

The paper can be accepted in current form

Response: Thank you for your suggestions that improve the quality of the research work. 

Reviewer 4 Report

I appreciate the review work. The answers given are quite accurate and complete. I recognize very significant improvements in the paper. I have only minor improvement suggestions:

- Decrease the size of Figure 11 to include the legend in the same page as the image.

- Avoid the expression “Use of most popular MCDM…” It is subjective.

Author Response

I appreciate the review work. The answers given are quite accurate and complete. I recognize very significant improvements in the paper. I have only minor improvement suggestions:

Response: Thank you for reviewing our paper.

- Decrease the size of Figure 11 to include the legend in the same page as the image.

Response:  Thanks for your suggestion, modified

Figure 11. Steps for Preference Ranking Organization Methods for Enrichment Evaluations (PROMETHEE II)

- Avoid the expression “Use of most popular MCDM…” It is subjective.

Response: Thanks for your suggestion, modified accordingly

The MCDM models such as TOPSIS, VIKOR, and PROMETHEE II, are having better decision-making ability for conflicting as well as non-conflicting criteria.